# *Listeria monocytogenes* cell-to-cell spread in epithelia is heterogeneous and dominated by rare pioneer bacteria

Fabian E Ortega[1], Elena F Koslover[1,2], Julie A Theriot[1,3†]*

[1]Department of Biochemistry, Howard Hughes Medical Institute, Stanford University, Stanford, United States; [2]Department of Physics, University of California San Diego, San Diego, United States; [3]Department of Microbiology and Immunology, Stanford University, Stanford, United States

**Abstract** *Listeria monocytogenes* hijacks host actin to promote its intracellular motility and intercellular spread. While *L. monocytogenes* virulence hinges on cell-to-cell spread, little is known about the dynamics of bacterial spread in epithelia at a population level. Here, we use live microscopy and statistical modeling to demonstrate that *L. monocytogenes* cell-to-cell spread proceeds anisotropically in an epithelial monolayer in culture. We show that boundaries of infection foci are irregular and dominated by rare pioneer bacteria that spread farther than the rest. We extend our quantitative model for bacterial spread to show that heterogeneous spreading behavior can improve the chances of creating a persistent *L. monocytogenes* infection in an actively extruding epithelium. Thus, our results indicate that *L. monocytogenes* cell-to-cell spread is heterogeneous, and that rare pioneer bacteria determine the frontier of infection foci and may promote bacterial infection persistence in dynamic epithelia.

**Editorial note:** This article has been through an editorial process in which the authors decide how to respond to the issues raised during peer review. The Reviewing Editor's assessment is that all the issues have been addressed (see decision letter).

DOI: https://doi.org/10.7554/eLife.40032.001

**\*For correspondence:**
jtheriot@uw.edu

**Present address:** [†]Department of Biology, Howard Hughes Medical Institute, University of Washington, Seattle, United States

**Competing interests:** The authors declare that no competing interests exist.

## Introduction

The widely studied foodborne pathogen *Listeria monocytogenes* has served as a model system to study cytoskeletal dynamics (*Theriot et al., 1992*; *Welch, 1998*), epithelial cell biology (*Pentecost et al., 2010*), and host-pathogen interactions (*Kocks et al., 1995*; *Mengaud et al., 1996*). This ubiquitous Gram-positive bacterium can invade and replicate within non-phagocytic cells and, importantly, use a form of actin-based motility to spread directly from the cytoplasm of an infected host cell into the cytoplasm of another host cell without exposure to the extracellular milieu (*Tilney and Portnoy, 1989*). This process, known as cell-to-cell spread, enables *L. monocytogenes* to breach and colonize the intestinal epithelium and to subsequently reach distant organs including the liver and brain in immunocompromised patients (*Ghosh et al., 2018*) and the placenta in pregnant women (*Faralla et al., 2016*). Indeed, compared to wild-type *L. monocytogenes*, mutant strains incapable of undergoing cell-to-cell spread are three orders of magnitude less virulent in murine models (*Domann et al., 1992*).

*L. monocytogenes* infections begin in the intestinal epithelium, a tissue made up of polarized epithelial cells connected to each other by cell-cell junctions (*Hartsock and Nelson, 2008*). *L. monocytogenes* preferentially adheres to and invades an epithelium at the tips of intestinal villi (*Pentecost et al., 2006*), where epithelial cells are actively extruded and shed (*Sancho et al., 2004*). Upon bacterial invasion, *L. monocytogenes* spreads to neighboring host cells, which can allow

**eLife digest** Eating food that has been contaminated with bacteria called *Listeria monocytogenes* can result in life-threatening infections. The bacteria first invade the epithelial cells that line the small intestine. After this, *L. monocytogenes* can move from one host cell to another, which allows the infection to reach other organs.

Most studies into how *L. monocytogenes* infections spread have focused either on how single bacterial cells move from one host cell to the next, or on how millions of bacteria damage host tissues. Little was known about the intermediate steps of an infection, where the bacteria start to colonize the small intestine.

To investigate, Ortega et al. recorded videos of *L. monocytogenes* spreading between epithelial cells grown on a glass coverslip, and developed computer simulations to try to reproduce how the bacteria spread. This revealed that the bacteria do not all move in the same way. Instead, less than 1% of the bacteria move in 'steps' that are up to 10 times longer than those taken by the others. Ortega et al. named these bacteria 'pioneers'.

Ortega et al. propose that the pioneers form long protrusions that allow them to spread directly from an infected cell to a non-neighboring cell. By taking these large steps, the pioneers may increase the chances that the bacteria will cause a long-lasting infection.

Future research will be needed to answer further questions about the pioneers. For example, how do the pioneer bacteria differ from the majority of bacterial cells? Would targeting anti-bacterial treatments at pioneers make it easier to treat infections? It also remains to be seen if other types of bacteria also show this pioneer behavior.

DOI: https://doi.org/10.7554/eLife.40032.002

bacteria to move away from the tip of a villus before the next host cell extrusion event terminates the infection. Therefore, understanding *L. monocytogenes* virulence requires a quantitative grasp of the spatiotemporal dynamics of cell-to-cell spread.

To initiate cell-to-cell spread, *L. monocytogenes* uses the protein ActA to polymerize actin at its surface and create an actin comet tail (*Pistor et al., 1994*). Actin polymerization generates a propulsive force that allows the bacterium to move within the host cytoplasm. Upon contact with the donor host cell membrane, the intracellular bacterium creates a protrusion that can extend into the cytoplasm of a recipient host cell (*Robbins et al., 1999*). Although cell-to-cell spread has been primarily studied as a mechanism of bacterial dissemination between adjacent host cells, it is well established that *L. monocytogenes* can create protrusions more than ten microns long (*Pust et al., 2005*), which could, in principle, mediate bacterial spread between two non-adjacent host cells. To complete cell-to-cell spread, the recipient cell engulfs the bacterium-containing protrusion, thus giving *L. monocytogenes* access to the recipient host cell's cytoplasm. After escaping the double-membrane vacuole, *L. monocytogenes* rebuilds the actin comet tail and restarts intracellular motility (*Gedde et al., 2000*).

Particular attention has been paid to bacterial and host cell proteins that mediate cell-to-cell spread. The bacterial protein internalin C helps *L. monocytogenes* to relax cortical tension and increase the rate of bacterial-mediated protrusion formation to promote spread (*Rajabian et al., 2009*). From the perspective of the host cell, it has been shown that TIM4 allows the host to sense bacterial-mediated membrane damage, which then triggers a repair mechanism that *L. monocytogenes* exploits to promote spread (*Czuczman et al., 2014*). The diaphanous-related formins (*Fattouh et al., 2015*) and members of the ERM protein family (*Pust et al., 2005*) have been shown to localize to bacteria-containing protrusions and inhibition of their activity decreases the efficiency of cell-to-cell spread. This cell biological approach has been useful in creating a mechanistic understanding of how individual spreading events occur. However, our larger scale understanding of how a population of bacteria spreads through tissue remains poorly developed.

Here, we combine live microscopy and statistical modeling to study the dynamics of a population of *L. monocytogenes* as it spreads through a polarized epithelial monolayer. We simulate cell-to-cell spread as an isotropic random walk because the movement of *L. monocytogenes* is directionally persistent over short distances but shows no preferred orientation over long distances. Our

experimental and computational results indicate that *L. monocytogenes* cell-to-cell spread includes a majority of local-spreading bacteria but is dominated by rare pioneers, which determine the shape of infection foci. Importantly, we find that pioneers alter the kinetics of spread in a way that might promote bacterial persistence in a dynamic epithelium where cells are actively extruded, as at the tip of an intestinal villus.

## Results

### *L. monocytogenes* spreads anisotropically through a polarized, confluent MDCK cell monolayer

To explore the dynamics of *L. monocytogenes* cell-to-cell spread in an epithelial monolayer, we developed a live video microscopy assay to track the progression of a bacterial infection over tens of hours. As a model host cell, we chose Madin-Darby canine kidney (MDCK) epithelial cells because they form polarized and homogeneous monolayers in culture (*Mays et al., 1995*) and have been widely used to study *L. monocytogenes* infection (*Robbins et al., 1999*; *Pentecost et al., 2006*; *Pentecost et al., 2010*). We infected confluent MDCK monolayers with a wild-type 10403 S *L. mono-cytogenes* strain that contains an mTagRFP open reading frame under the *actA* promoter, which becomes transcriptionally active when the bacterium enters the host cell cytosol (*Moors et al., 1999*; *Zeldovich et al., 2011*). We then imaged the progression of the infection as described in Materials and Methods. The presence of gentamicin, a bacteriostatic antibiotic that cannot cross the host cell plasma membrane (*Portnoy et al., 1988*), during live imaging ensured that only intracellular bacteria contributed to the growth and spread of the infection focus. Starting at approximately 6 hr post-infection, the earliest time point at which we could detect mTagRFP protein expression, we imaged bacterial foci for up to 22 hr post-infection (first three panels of *Figure 1A*, and *Video 1*).

Given that bacterial invasion of a polarized MDCK monolayer is a rare event (*Pentecost et al., 2006*), each infection focus most likely began with a single bacterium entering a host cell's cytosol. Due to the clonal nature of the replicating bacteria, and the homogeneity of the host monolayer, we were surprised to find behavioral heterogeneity within the bacterial population; the edges of the boundary of the infection focus, determined by the smallest boundary that completely encloses all bacteria, were dominated by a small number of bacteria that spread farther than the rest (*Figure 1A*, white arrows in third panel). Indeed, this was a common phenomenon that could be observed in most infection foci. Although each focus may have started out roughly circular, far-spreading bacteria, which we refer to as 'pioneers', nearly always created irregular boundaries by the end of the experiment (*Figure 1B*).

Despite boundary irregularity, intracellular bacterial replication was approximately exponential (*Figure 1C*), and the growth rate could be modeled with a one-term exponential function (*Figure 1—figure supplement 1A*) with an average doubling time of approximately 180 min (*Figure 1—figure supplement 1C*). This doubling time is comparable to what has been previously reported in other epithelial host cell types using gentamicin protection assays (*Gaillard et al., 1987*). Between 360 and 960 min post-infection, the mean squared displacement (MSD) of the bacterial positions (defined here as the second moment of the fluorescence intensity distribution) appeared linear (*Figure 1C*), which is consistent with a random walk (*Berg, 1993*). However, the slope of the MSD was not always constant, but instead increased with time (*Figure 1—figure supplement 1B*), which is consistent with the appearance of fast-spreading organisms within a migrating population (*Shigesada et al., 1995*). We found no correlation between bacterial growth rate and MSD (*Figure 1—figure supplement 1C*), enabling us to treat these two parameters as independent in the quantitative model described below.

### Stochastic simulations of cell-to-cell spread via random walks are inconsistent with observed shapes of infection foci

What then is the expected range of shapes resulting from the random movement and exponential growth seen in *L. monocytogenes* cell-to-cell spread? From the literature, it is expected that, when starting from a point source, random movement and growth should yield isotropic shapes (*Holmes et al., 1994*). To formalize this null hypothesis, we solved the reaction-diffusion equation (*Equation 1*):

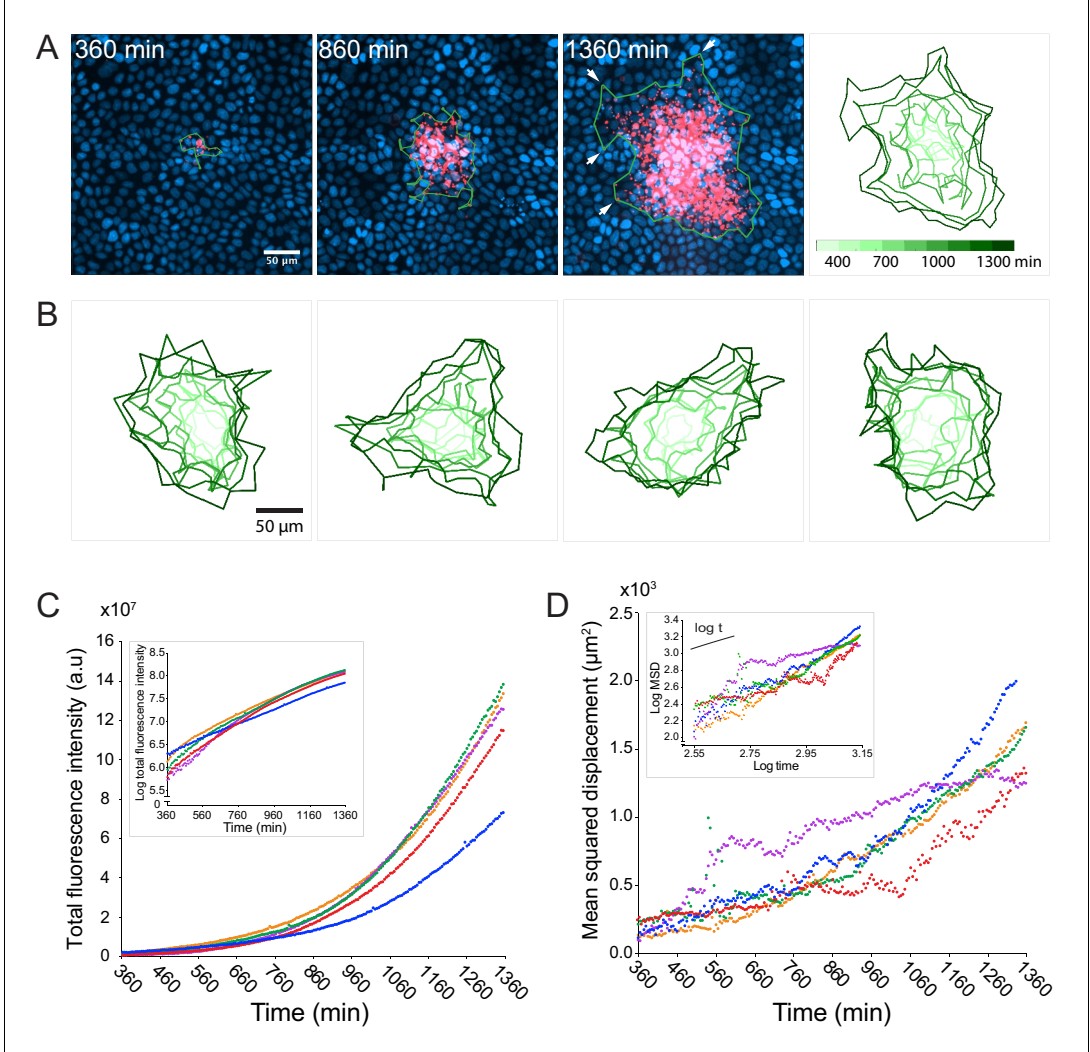

**Figure 1.** *L. monocytogenes* spreads anisotropically through a polarized, confluent MDCK cell monolayer. (**A**) On the left three panels, micrographs show nuclei (Hoechst, blue) and intracellular bacteria (mTagRFP, red) at three different time points post-infection. Green boundaries fully enclose all bacteria. On the fourth panel, boundaries (shades of green) depict the progression of the infection focus boundaries at nine evenly-spaced time points (see colorbar). (**B**) Examples of four different foci boundaries through time. (**C**) Quantification of total bacterial fluorescence intensity as a function of time for five different foci. Semi-log plot in inset. (**D**) Quantification of mean squared displacement (MSD) as a function of time for five different foci. Log-log plot in inset where short solid line indicates linear scaling. For C and D, each focus is represented by a different color.

DOI: https://doi.org/10.7554/eLife.40032.003

The following source data and figure supplement are available for figure 1:

**Source data 1.** This spreadsheet contains the total fluorescence intensity (a.u.), mean squared displacement ($\mu m^2$), doubling time (min), and diffusion coefficient ($\mu m^2$/min) data used to generate the graphs in *Figure 1C and D*, and in *Figure 1—figure supplement 1A, B and C*.

DOI: https://doi.org/10.7554/eLife.40032.005

**Figure supplement 1.** Doubling times (min) and effective diffusion coefficients ($\mu m^2$/min) for live microscopy data are uncorrelated.

DOI: https://doi.org/10.7554/eLife.40032.004

$$\frac{\partial \phi}{\partial t} = D \frac{\partial^2 \phi}{\partial r^2} + k\phi \qquad (1)$$

where Φ represents the bacterial concentration as a function of position and time, *t* refers to time, *r* refers to the position of the bacteria in polar coordinates, *D* is the effective diffusion coefficient, and *k* is the exponential growth rate. Variations of this partial differential equation have been used to model dynamic biological processes such as morphogen pattern formation (*Gordon et al., 2011*)

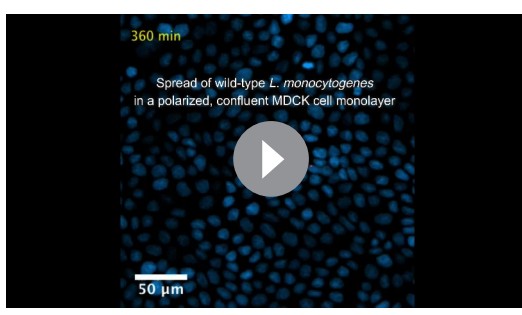

**Video 1.** Spread of wild-type *L. monocytogenes* in a polarized, confluent MDCK cell monolayer. Time-lapse microscopy data showing intracellular wild-type bacteria (mTagRFP, red) and host cell nuclei (Hoechst, blue). Images were collected every 5 min for a total of 1000 min. Time shown in minutes post-infection.
DOI: https://doi.org/10.7554/eLife.40032.006

and animal migration (*Skellam, 1951*). Because *Equation 1* is radially isotropic, its solutions correspond to circular infection foci that grow in size and intensity over time (*Figure 2A* and *Video 2*). Such a continuum model cannot account for the experimentally observed heterogeneous focus shapes.

It is important to note that treating the bacterial concentration Φ as a continuous variable constitutes a mean-field approximation, which is valid only in the limit of high bacterial counts, and which neglects correlations in the positions of individual bacteria. However, because each bacterium behaves as a discrete entity and because the number of bacteria at the start of each infection focus is very small, the mean-field model breaks down in describing the shape of individual foci. Stochastic variation in the trajectories of individual bacteria, amplified by exponential growth, could in principle lead to more irregular focus shapes such as those observed in *Figure 1A–B*. We thus turned to simulations with finite numbers of discrete bacterial agents to examine the effect of such stochastic fluctuations.

Agent-based simulations have been used to study discrete biological phenomena such as the spread of infectious endemic agents throughout populations (*Juher et al., 2009*) and the diversification of lymphocyte antigen-receptor repertoires (*Castiglione, 2011*). The benefit of using this method to model *L. monocytogenes* cell-to-cell spread is that it allows simulation of individual bacteria as discrete particles and avoids the continuum assumption imposed by *Equation 1*. We match the simulation run time to experimental conditions, proceeding until $10^5$ bacteria are accumulated. The primary goal was to determine whether fluctuations arising from random trajectory sampling were sufficient to account for the observed boundary anisotropy.

In these simulations, individual bacteria execute an isotropic random walk in two dimensions (*Video 3*), with the step in each dimension selected from a normal distribution with mean zero and variance 2DΔt where Δt is the simulation time-step. Each bacterium replicates at a preset time interval after its initial birth, resulting in an overall replication rate k (*Figure 2B* and *Video 4*). The MSD and total counts of simulated bacteria accurately reflect the input parameters of diffusivity D and replication rate k (*Figure 2—figure supplement 1*). As expected, the speed of the infection focus boundary, defined as the square root of the area of the boundary, approaches the theoretical limit of 2 times the square root of Dk (*Liebhold and Tobin, 2008*) at long times (Materials and Methods; *Figure 2—figure supplement 2*).

While not perfectly isotropic, the stochastic simulations generated foci that were approximately circular and thus differed significantly from the experimental foci (*Figure 2C*). To quantify the circularity of the experimental and simulated foci, we calculated the ratio of the area of a focus over the area of the smallest circle that fully encloses the focus (*Figure 2D*). For a perfect circle, this metric would be equal to 1, and for a square, this metric would be equal to 2/π (*Zheng and Hryciw, 2015*). Importantly, this metric is not dependent on the focus size (*Figure 2—figure supplement 3*). For all measurements, simulated foci were convolved with the point spread function of individual bacterial cells to match the empirically determined resolution of our microscope system, so that simulation outputs could be directly compared to experimental observations (Materials and Methods). The data showed that simulated foci are substantially more circular than experimental foci (*Figure 2E*).

It is known that intracellular *L. monocytogenes* does not undergo truly uncorrelated random walks as was assumed in our simulations. Instead, intracellular *L. monocytogenes* motility, aided by ActA-dependent actin comet tails, exhibits directional persistence over time-scales of a few minutes (*Lacayo and Theriot, 2004*; *Soo and Theriot, 2005*). In addition, our initial simulations ignored the presence of host cell boundaries, which *L. monocytogenes* encounters as they spread from cell to cell. In fact, it has been shown that *L. monocytogenes* can ricochet off MDCK host cell boundaries at

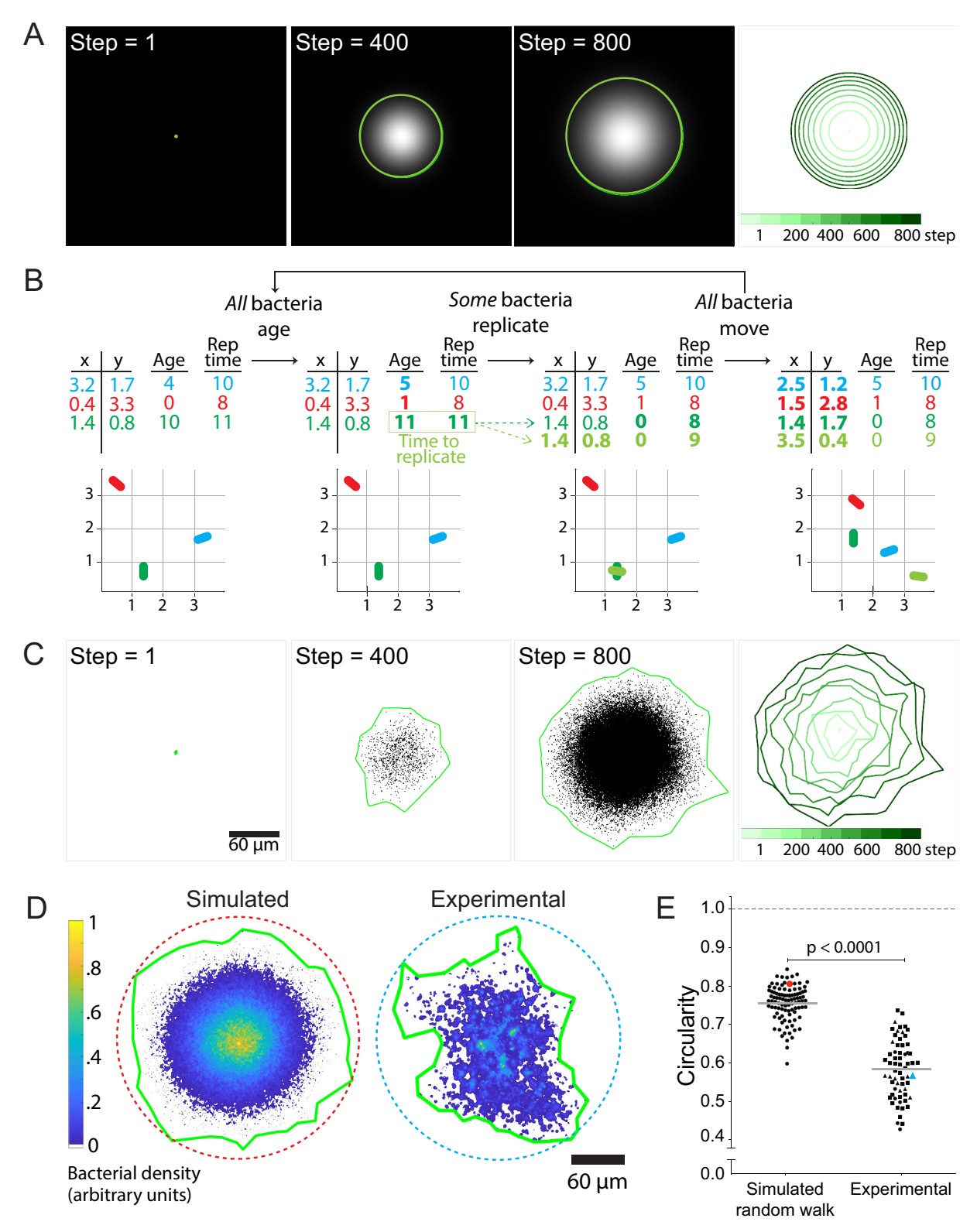

**Figure 2.** Stochastic simulations of cell-to-cell spread via random walks are inconsistent with observed shape of infection foci. (**A**) On the left three panels, images depict the solution of the reaction-diffusion equation at three different time steps. Green boundaries enclose the brightest 90% of pixels. On the fourth panel, boundaries (shades of green) depict the progression of the shape of the differential equation solution at nine evenly-spaced time steps (see colorbar). (**B**) Schematic of stochastic simulation. Each bacterium, depicted by a different color, is represented by a location in

*Figure 2 continued on next page*

*Figure 2 continued*

Cartesian coordinates, age, and replication time. A bacterium replicates when its age catches up with its replication time. Bacteria take steps that are normally distributed and scaled by an effective diffusion coefficient D, and they replicate according to a replication rate k. (**C**) The left three panels depict three time steps of a random walk stochastic simulation where D = 1 and k = 1. Each simulated bacterium is depicted by a data point. Green boundaries fully enclose all data points. On the fourth panel, boundaries (shades of green) depict the progression of a simulated focus boundaries at nine evenly-spaced time steps. (**D**) Representative simulated focus convolved with the *L. monocytogenes* point spread function (left). Representative experimental focus (right). Green boundaries fully enclose all bacteria. Circular dashed lines represent the smallest circles that fully enclose green boundaries. Colorbar indicates bacterial density. (**E**) Data quantifying infection foci circularity of simulated data versus experimental data. For experimental data, each shape depicts an independent experiment. Horizontal bars indicate the mean. p-Value was calculated with the non-parametric Wilcoxon rank sum test. The red and cyan data points correspond to the simulated and experimental foci, respectively, from panel D.

DOI: https://doi.org/10.7554/eLife.40032.007

The following source data and figure supplements are available for figure 2:

**Source data 1.** This spreadsheet contains circularity data used to generate graphs in *Figure E*, in *Figure 2—figure supplement 2*, in *Figure 2—figure supplement 3B*, and in *Figure 2—figure supplement 4A and B*.

DOI: https://doi.org/10.7554/eLife.40032.012

**Figure supplement 1.** Stochastic model accurately simulates a random walk.

DOI: https://doi.org/10.7554/eLife.40032.008

**Figure supplement 2.** Radial speed of the frontier of infection foci approaches the square root of 2Dk as time increases.

DOI: https://doi.org/10.7554/eLife.40032.009

**Figure supplement 3.** Circularity does not depend on the size of the simulated infection focus.

DOI: https://doi.org/10.7554/eLife.40032.010

**Figure supplement 4.** Host cell boundaries and motility persistence do not affect simulated infection focus circularity.

DOI: https://doi.org/10.7554/eLife.40032.011

a frequency dependent on monolayer age (*Robbins et al., 1999*). To test the possibility that bacterial motility persistence and the presence of host cell boundaries could affect the circularity of the simulated foci, we updated our simulations to include both of these effects (Materials and Methods, and *Video 5*). We found that neither of these two conditions affects circularity significantly; specifically, foci simulated with these features are only about 3% less circular than foci simulated by a random walk alone (*Figure 2—figure supplement 4A*). Overall, changing the probability with which the bacteria cross host cell boundaries had a minimal effect on the circularity of the simulated foci (*Figure 2—figure supplement 4B*).

Taken together, our experimental and simulated data show that *L. monocytogenes* cell-to-cell spread cannot be modeled with only a random walk and exponential growth, and that the presence of host cell boundaries and the persistence of bacterial motility do not have a significant effect on the circularity of infection foci. We therefore decided to look more closely at the influence and significance of pioneer bacteria.

## Allowing simulated bacteria to interconvert between pioneer and non-pioneer behavior recapitulates the non-circular phenotype of experimental foci

Pioneer bacteria, which spread unusually far compared to the overall bacterial population (*Figure 3A* and *Video 6*), have the potential to substantially alter the shape and isotropy of infection foci. For our experiments performed in the presence of extracellular gentamicin, *L. monocytogenes* only replicates in a host cell's cytoplasm. We also know from direct observation that at least one round of division must take place before bacteria can resume actin-based motility in the recipient cell (*Robbins et al., 1999*). Therefore, the simplest explanation

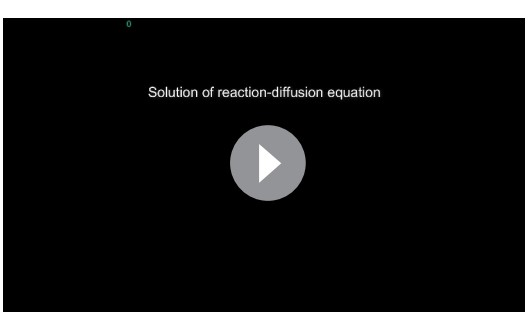

**Video 2.** Solution of reaction-diffusion equation. Solution as defined in *Equation 2*. Effective diffusion coefficient and replication rate were each set to 1. Time steps were set to 0.01.

DOI: https://doi.org/10.7554/eLife.40032.013

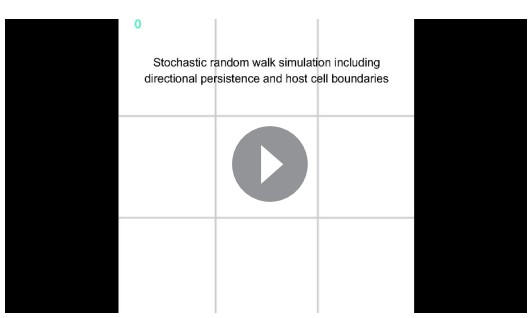

**Video 3.** Initial steps of stochastic random walk simulation. First 425 steps of a stochastic simulation that starts with a single bacterium and ends when 100 bacteria have accumulated. Effective diffusion coefficient and replication rate equal 1.
DOI: https://doi.org/10.7554/eLife.40032.014

**Video 4.** Full stochastic random walk simulation. Stochastic simulation that starts with 25 bacteria and ends when $10^5$ bacteria have accumulated. Effective diffusion coefficient and replication rate equal 1.
DOI: https://doi.org/10.7554/eLife.40032.015

consistent with our direct observation of events such as the one shown in **Figure 3A**, is that the bacterium travels from a donor cell directly to a non-adjacent recipient through a long protrusion (approximately 15–20 µm in the example shown). Once inside the cytoplasm of this non-adjacent recipient cell, the bacterium replicates. Importantly, this bacterium can reach a non-adjacent recipient host cell in less than 30 min even though it takes an MDCK cell approximately 45 min to complete the process of taking up a bacterium-containing protrusion (**Robbins et al., 1999**). In contrast, non-pioneer bacteria typically move about 1–2 µm in 5-min intervals in our assay (**Video 6**).

For a few of these pioneer events, the pioneer bacterium went transiently out of focus in our widefield imaging setup, consistent with the possibility that this long protrusion extended above the apical surface of the monolayer (**Video 7**). However, such long protrusions reaching non-adjacent cells could in principle also extend beneath the basal surface of the monolayer or indeed even between cell-cell junctions. For MDCK cells, the tight junctions which presumably would occlude lateral extension of long protrusions between neighboring cells only comprise the top 5–10% of the lateral face of the cells in culture (**Nelson and Veshnock, 1986**), so there is ample space for long protrusions to extend between cells in the monolayer prior to protrusion uptake by a non-adjacent recipient host.

Pioneers, which appear to determine the frontier of the infection focus boundary (**Figure 1A–B**), can be incorporated into the stochastic simulation by allowing bacteria to sample from an alternate distribution of step sizes. For simplicity, we thus include pioneers in our model by allowing all bacteria to move in a purely diffusive fashion, with either a slow (non-pioneer) diffusivity $D_{slow}$ or a fast (pioneer) diffusivity $D_{fast}$. Pioneer behavior in the simulations is then characterized by the ratio of $D_{fast}/D_{slow}$ (i.e. how much further pioneers spread as compared to non-pioneers) and the probability with which a bacterium becomes a pioneer. When a bacterium replicates, each daughter has a probability P of spreading according to $D_{fast}$ and probability 1–P of spreading according to $D_{slow}$ (**Video 8**). We assume the assignment of each individual bacterium as either a pioneer or non-pioneer persists until a bacterium's next replication event.

We first simulated cell-to-cell spread by setting the probability of becoming a pioneer to 0.10 and the $D_{fast}/D_{slow}$ ratio to 100. These are

**Video 5.** Stochastic random walk simulation including directional persistence and host cell boundaries. Directional persistence equals 0.3 and the probability of crossing a host cell boundary equals 0.10. Host cell boundaries are drawn in gray. The trailing points behind a bacterium represent the last ten positions of that bacterium. Effective diffusion coefficient and replication rate equal 1.
DOI: https://doi.org/10.7554/eLife.40032.016

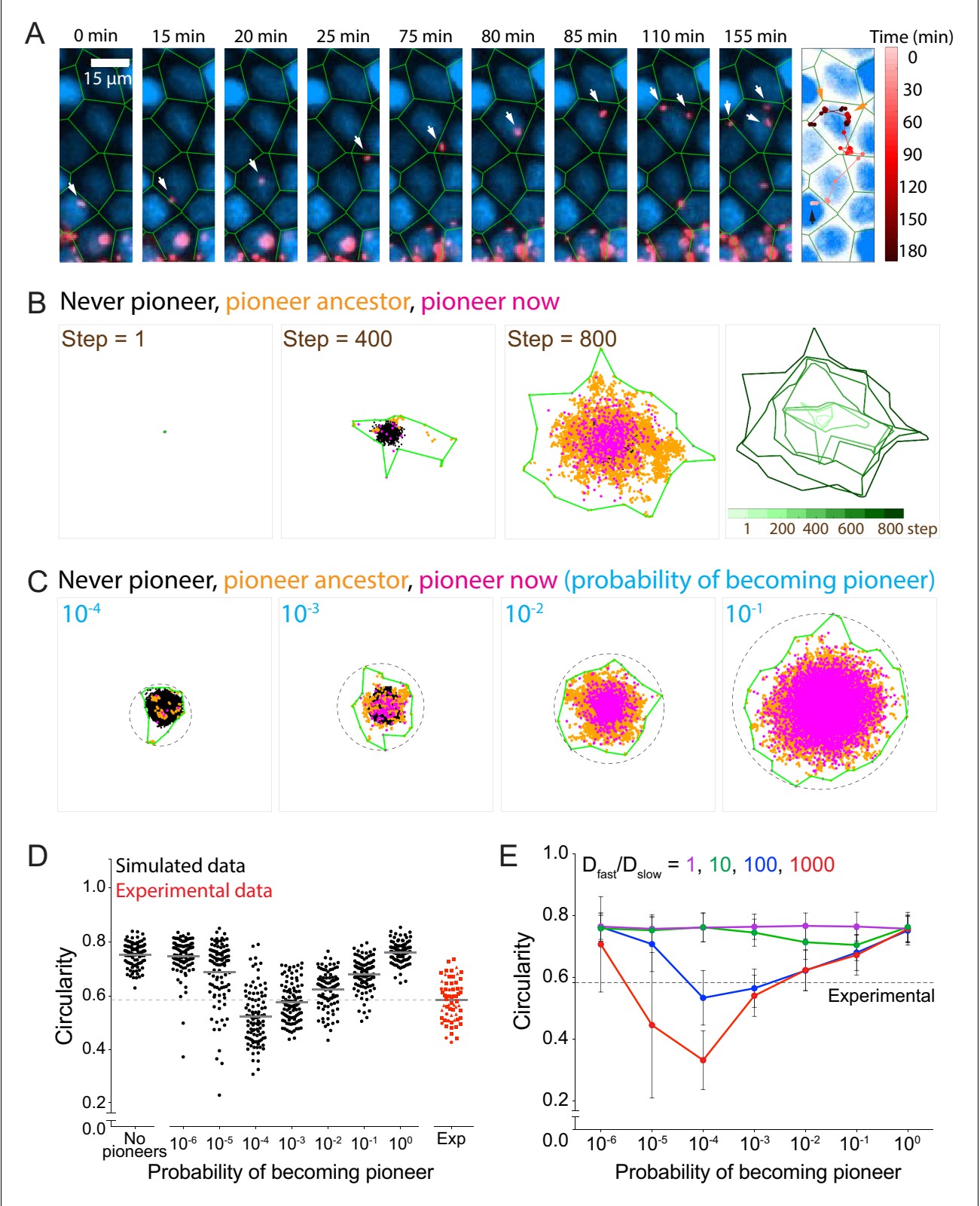

**Figure 3.** Allowing simulated bacteria to interconvert between pioneer and non-pioneer behavior recapitulates the non-circular phenotype of experimental foci. (A) On the left nine panels, micrographs show nuclei (Hoechst, blue) and intracellular bacteria (mTagRFP, red). Green lines, which serve as an approximation of host cell boundaries, depict the Voronoi tessellation of the centroids of the host nuclei. White arrows track a single pioneer and its two daughters through time. Start time (leftmost panel) refers to 1255 min post-infection. The right panel shows nuclei (Hoechst, blue)

*Figure 3 continued on next page*

*Figure 3 continued*

and the pioneer path throughout 180 min. Dots indicate the position of the pioneer at a given time point. Shades of red depict progression of time. Black arrow indicates start position. Orange arrows indicate the location and time of two bacterial replication events. (B) The left three panels depict three time steps of a stochastic simulation where $D_{slow}$ = 1, $D_{fast}$ = 100, P = 0.10, and k = 1. Each simulated bacterium is depicted by a data point. Green boundaries fully enclose all data points. On the fourth panel, boundaries (shades of green) depict the progression of a simulated focus boundaries at nine evenly-spaced time steps. (C) Images depicting $10^5$ simulated bacteria at step 800 of stochastic simulations where $D_{slow}$ = 1, $D_{fast}$ = 100, k = 1, and probability of becoming a pioneer is depicted in cyan. Green boundaries fully enclose all data points. Circular dashed lines represent the smallest circles that fully enclose the green boundaries. (D) Data quantifying the circularity of experimental (red) and simulated (black) foci at step 800, which, normalized by the replication rate (0.006 $min^{-1}$), is equivalent to approximately 1360 min of experimental time. For experimental data, each shape depicts an independent experiment. Dashed line at circularity of 0.58 refers to the mean of the experimental circularity. Horizontal bars indicate the mean for each condition. (E) Data quantifying circularity of simulated infection foci as a function of probability of becoming a pioneer. Each data point represents the average of 100 independent simulations. Vertical bars represent the standard deviation. Each color represents a different value of $D_{fast}/D_{slow}$ ratio. Dashed line around 0.58 refers to the mean of the experimental circularity. For all simulations, replication rate k equals 1.

DOI: https://doi.org/10.7554/eLife.40032.017

The following source data and figure supplements are available for figure 3:

**Source data 1.** This spreadsheet contains circularity data used to generate graphs in *Figure 3D and E*, in *Figure 3—figure supplement 1B and C*, and in *Figure 2—figure supplement 2A and B*.

DOI: https://doi.org/10.7554/eLife.40032.020

**Figure supplement 1.** Allowing simulated bacteria to interconvert between pioneer and non-pioneer behavior significantly affects the circularity of the infection focus.

DOI: https://doi.org/10.7554/eLife.40032.018

**Figure supplement 2.** As time increases, focus circularity increases in both simulated and experimental data.

DOI: https://doi.org/10.7554/eLife.40032.019

reasonable parameters because (1) a relatively small number of bacteria spread much farther than the rest throughout a live microscopy assay, and (2) an effective diffusion coefficient ratio of 100 translates to pioneer steps that are an order of magnitude longer than non-pioneer steps, which is consistent with what we observe experimentally (*Figure 3A*). As expected, the presence of pioneers caused the simulated boundaries to become anisotropic, particularly in the early steps of the simulation (*Figure 3B*). Additionally, these simulations recapitulated the increase in the MSD slope during the later time points of the experimental data (*Figure 3—figure supplement 1A*). The transition to a larger MSD occurs at a time when the bacterial population stabilizes to contain a larger fraction of pioneers. A probability of 0.10 allowed, on average, approximately 75% of bacteria to have a pioneer ancestor or be pioneers themselves by the end of the simulation (*Figure 3—figure supplement 1B*). It is likely that the approach towards a pioneer majority explains why simulated foci boundaries tended to be more anisotropic during earlier steps of the simulation, and why they became more circular as simulation time increased (*Figure 3—figure supplement 1C*). Circularity of simulated foci would sometimes drop precipitously if the probability of becoming a pioneer was less than 0.001 (*Figure 3—figure supplement 2A*). We also observed a general time-dependent increase in infection focus circularity in experimental data, which also sometimes exhibited rapid decreases in infection focus circularity (*Figure 3—figure supplement 2B*). The observed kinetics of changes in circularity over time for both the experimental data and the simulations are consistent with the proposition that the overall focus size and shape become more strongly dominated by the pioneers at later time points, as also illustrated by the transition in MSD slope described above. Overall, less circular foci shapes were observed when the pioneer probability was low enough so that only a few bacteria

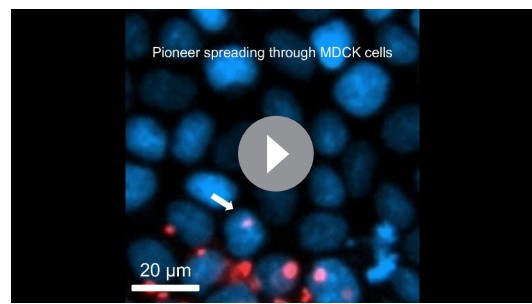

**Video 6.** Pioneer spreading through MDCK cells. Single pioneer spreading through a polarized monolayer of MDCK cells. Images were collected every 5 min. White arrow indicates the initial position of pioneer bacterium.

DOI: https://doi.org/10.7554/eLife.40032.021

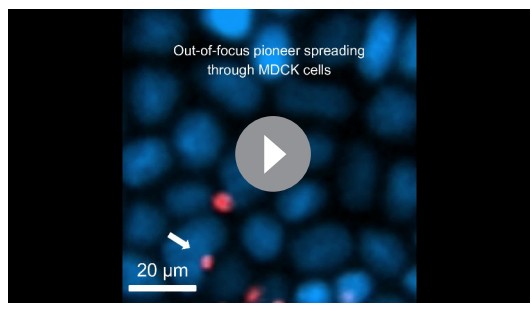

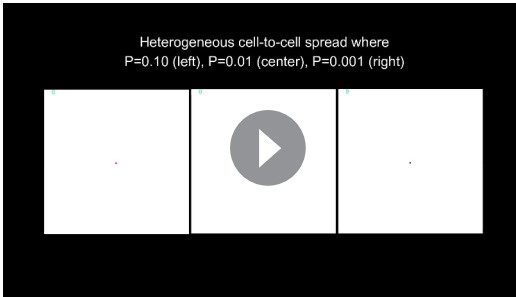

**Video 7.** Out-of-focus pioneer spreading through MDCK cells. Single pioneer briefly goes out of focus as it spreads through a polarized monolayer of MDCK cells. Images were collected every 5 min. White arrow indicates the initial position of pioneer bacterium.
DOI: https://doi.org/10.7554/eLife.40032.022

**Video 8.** Heterogeneous cell-to-cell spread where P=0.10 (left), P=0.01 (center), and P=0.001 (right). Stochastic simulations where $D_{slow} = 1$, $D_{fast} = 100$, and $k = 1$. Each simulation starts with 25 bacteria and ends when $10^5$ bacteria have accumulated. Colors depict whether a bacterium has never been a pioneer (black), has a pioneer ancestor (orange), or is currently a pioneer (magenta). Green boundaries fully enclose all bacteria.
DOI: https://doi.org/10.7554/eLife.40032.023

in each focus exhibited pioneer behavior (*Figure 3C*).

We confirmed these findings quantitatively and showed that experimental circularity is equivalent to the circularity seen for simulations with pioneer probabilities of $10^{-3}$ and $10^{-2}$ (*Figure 3D*), which suggests that during our cell-to-cell spread experimental assay, approximately 1.4% to 12% of bacteria have pioneer ancestors or are pioneers themselves (*Figure 3—figure supplement 1B*). We note that these results are dependent on the total simulation time, as higher overall bacterial counts (longer simulation times) result in more circular foci for the same value of pioneer probability. We also tested the effect on circularity of changing the ratio of $D_{fast}/D_{slow}$. As expected, a larger ratio, that is pioneers taking longer steps than non-pioneers, decreases circularity significantly more than a smaller ratio (*Figure 3E*).

Together our findings suggest that *L. monocytogenes* cell-to-cell spread is consistent with individual bacteria having a low but non-zero probability of becoming pioneers, while the majority of the bacteria spread locally. We refer to this form of *L. monocytogenes* dissemination as heterogeneous cell-to-cell spread following terminology from the ecological study of animal dispersion (*Shigesada et al., 1986*). We next investigated whether *L. monocytogenes* entering straight long protrusions could form the basis of heterogeneous spread.

## Decreasing the persistence of bacterial motility leads to more circular infection foci

During *L. monocytogenes* cell-to-cell spread, it is known that intracellular bacteria create protrusions that can be taken up by a recipient cell directly adjacent to the donor cell. In this case, donor and recipient cells are connected to one another by the protein-protein interactions of constituents of adherens and tight junctions (*Hartsock and Nelson, 2008*). However, to explain the pioneer phenomenon, we propose that a few bacteria will create longer protrusions that will allow them to reach a more distant recipient cell that is not adjacent to the donor cell. In other words, this spreading event takes place between two cells that do not form junctions directly with each other (*Figure 4A*). This is a reasonable hypothesis because *L. monocytogenes* can form long protrusions that are tens of microns in length, sufficient to allow them to reach non-adjacent host cells (*Pust et al., 2005*). *L. monocytogenes*' ability to create long, pioneer-containing protrusions thus would be critical for the complex, non-circular boundaries observed in experimental data.

To test this model, we infected confluent MDCK cell monolayers with either wild-type *L. monocytogenes* or an *L. monocytogenes* strain where the proline residues in three proline-rich regions of the ActA protein have been mutated to glycine (*Skoble et al., 2001*). This mutant, known as the glycine-rich repeat (GRR) mutant, is less persistent than wild-type bacteria, which means that it loses its original direction more quickly than wild-type bacteria. The GRR mutant is also characterized by two-fold shorter actin comet tails (*Auerbuch et al., 2003*). These characteristics make the GRR

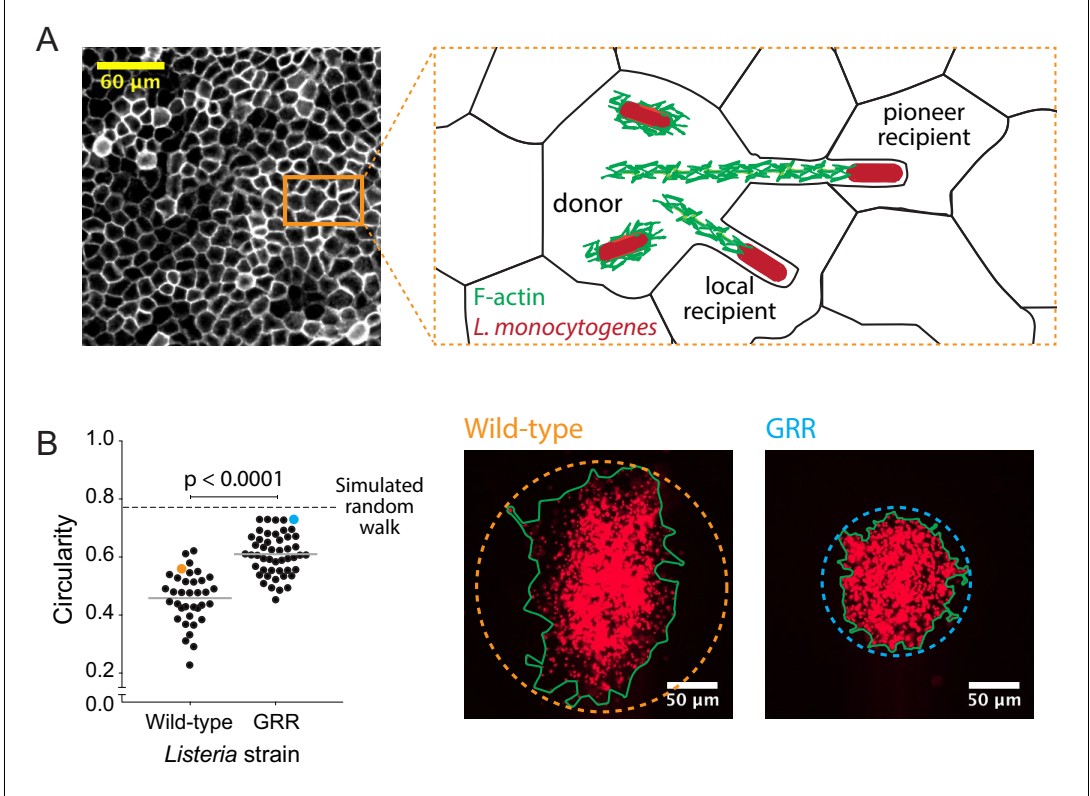

**Figure 4.** Decreasing the rate of bacterial protrusion formation leads to more circular infection foci. (**A**) Micrograph showing phalloidin staining of MDCK cell monolayer (left), and a cartoon representation of host cell boundaries based on phalloidin staining and *L. monocytogenes* spreading from cell to cell. (**B**) Data quantifying the circularity of foci generated by wild-type and GRR *L. monocytogenes.* Horizontal bars indicate the mean. p-value was calculated using the non-parametric Wilcoxon rank sum test. The orange and cyan data points correspond to the foci on the right. Green boundaries fully enclose all bacteria. Circular dashed lines represent the smallest circles that fully enclose green boundaries.
DOI: https://doi.org/10.7554/eLife.40032.024

The following source data and figure supplement are available for figure 4:

**Source data 1.** This spreadsheet contains circularity data used to generate graphs in *Figure 4B* and in *Figure 4—figure supplement 1*.
DOI: https://doi.org/10.7554/eLife.40032.026
**Figure supplement 1.** Changing the value of $D_{fast}/D_{slow}$ significantly affects the circularity of simulated foci; changing directional persistence does not.
DOI: https://doi.org/10.7554/eLife.40032.025

mutant likely to enter protrusions at a lower frequency than wild-type bacteria and to form protrusions that are less straight. Upon quantifying the circularity of GRR foci, we found that they were significantly more circular than foci created by wild-type *L. monocytogenes* (*Figure 4B*). This was likely a consequence of a decrease in the probability of forming long, straight protrusions, which then decreased the probability of bacteria exhibiting pioneer behavior. Indeed, changing the directional persistence in simulations including pioneers had little effect in circularity (*Figure 4—figure supplement 1*), thus supporting the idea that pioneer behavior, that is making long straight protrusions, has a stronger effect in circularity than intracellular directional persistence. Because pioneer behavior in wild-type bacteria is expected to occur quite rarely, decreasing the pioneer probability still further should result in more circular infection foci as very few pioneering events occur over the observation period (*Figure 3D*).

Our findings suggest that *L. monocytogenes* cell-to-cell spread is heterogeneous as it proceeds via local non-pioneers and far-spreading pioneers, each of which can be modeled with a random walk. In addition, we have shown that pioneer behavior is probably based on *L. monocytogenes'* ability to spread directly to non-adjacent host cells via long extracellular protrusions. However, because of the intrinsic limitations of our wide-field imaging methodology, we cannot tell whether

these long, straight protrusions extend above, below, or between host cells as they are reaching their destination.

## Simulations predict that heterogeneous spread increases the chance of a persistent *Listeria monocytogenes* infection in the intestinal epithelium

In considering the possible biological significance of pioneer behavior, we next asked whether heterogeneous cell-to-cell spread would promote *L. monocytogenes* intracellular survival and growth in a more physiological setting. To answer this question, we updated our simulations to more closely mimic the physiology of the tip of an intestinal villus by including host cell extrusion events, which could terminate bacterial infections *in vivo* (*Figure 5A*). Given *L. monocytogenes*' ability to spread away from an actively extruding villus tip, the rate of host cell extrusion and the rate of *L. monocytogenes* cell-to-cell spread together determine the fate of an intestinal infection. In the updated simulations, after a pre-determined period of time, a circular host cell at the center of the simulated monolayer is removed (extruded), taking with it the bacteria found inside. The monolayer then contracts to replace the extruded host cell and moves all other bacteria radially inward (*Figure 5B*). As before, simulated bacteria spread via a random walk and replicate exponentially. The simulation keeps track of both the number of bacteria in the monolayer and the number of bacteria that have been extruded.

Unlike previous simulations, which we terminated at the point at which $10^5$ total bacteria had accumulated, we ended host cell extrusion simulations in one of three ways: (1) no bacteria left in the monolayer, called bacterial clearance (*Video 9*, left); (2) too many bacteria, for example $10^5$, have accumulated in the monolayer, called uncontrolled growth (*Video 9*, right); (3) the number of bacteria extruded from the monolayer has reached a pre-determined threshold, for example $2 \times 10^5$, without accumulating too many bacteria in the monolayer. This third outcome, which we term a stable steady state, is equivalent to a persistent infection that allows *L. monocytogenes* to actively replicate and spread in the epithelium while being kept in check by the animal's host cell extrusion (*Video 9*, center). Stable steady state does not harm the host, and it allows *L. monocytogenes* to exit the animal via feces and infect other animals, which benefits the pathogen (*Begley et al., 2005*; *Roldgaard et al., 2009*).

To learn about the relationship between the rate of *L. monocytogenes* cell-to-cell spread and the rate of host cell extrusion, we ran random walk simulations and varied both the effective diffusion coefficient (D) and the host cell extrusion period (E). Specifically, we ran 100 independent simulations for each combination of D and E and quantified the outcomes. We found that small values of E, indicative of an actively extruding monolayer, favored bacterial clearance, and that large values of E, indicative of a more quiescent monolayer, favored uncontrolled growth, as expected. Similarly, small values of D favored bacterial clearance and large values of D favored uncontrolled growth. Stable steady state, on the other hand, was only reached by a narrow set of intermediate values of D and E (*Figure 5C*), corresponding to parameters where the rate of bacterial removal by extrusion was precisely balanced by the rate of replication (as derived in Materials and Methods).

We were next interested in asking whether heterogeneous cell-to-cell spread would increase the chances that *L. monocytogenes* could attain a stable steady state in an actively extruding epithelium. We first chose conditions that produced 100% bacterial clearance outcomes in the case of a random walk, by setting D = 2 and E = 0.15 (*Figure 5—figure supplement 1*). Next, to simulate heterogeneous spread, we set $D_{slow}$ = D, kept E the same, varied the value of $D_{fast}$, and set P = 0.01, where p is the probability of becoming a pioneer at the time of birth. Interestingly, values of $D_{fast}$ that were 60- to 90-fold higher than $D_{slow}$ allowed *L. monocytogenes* to reach a stable steady state (*Figure 5D*). A $D_{fast}/D_{slow}$ ratio in this range translates to pioneer bacteria taking steps 7.4- to 9.5-fold longer as compared to non-pioneer bacteria, which is consistent with our experimental observations (*Figure 3A*). In addition, many $D_{fast}/D_{slow}$ ratios, for several host cell extrusion periods, allowed *L. monocytogenes* to attain a stable steady state (*Figure 5—figure supplement 2*).

Together, our findings argue that *L. monocytogenes* heterogeneous cell-to-cell spread improves the chances of the pathogen reaching a stable steady state *in vivo* as compared to bacteria spreading via a random walk alone. The combination of these outcomes would prevent damage to the host animal tissue, facilitate bacterial dissemination to other host animals, and allow *L. monocytogenes* to thrive in the actively extruding and ever-changing intestinal epithelium.

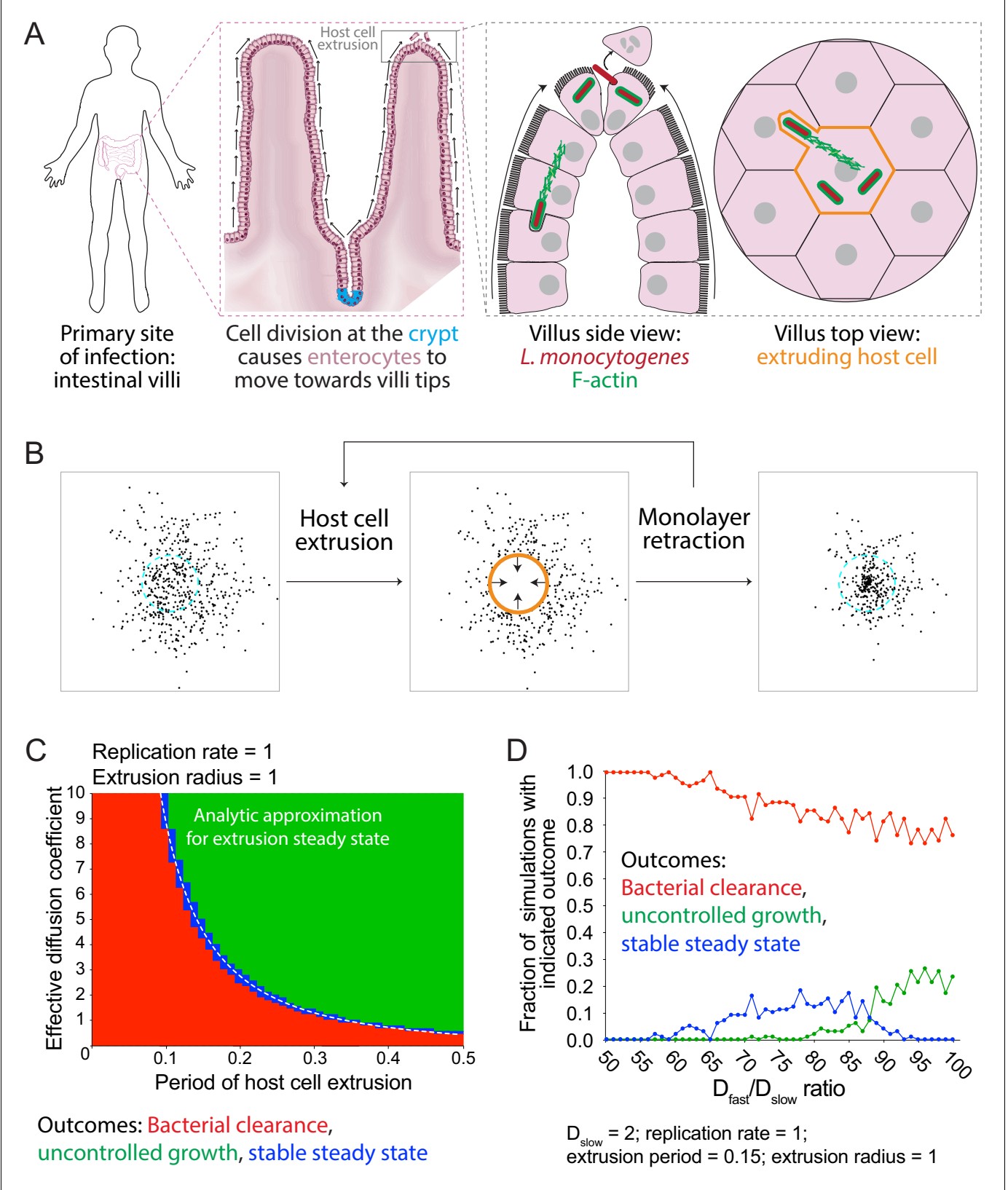

**Figure 5.** Simulations predict that heterogeneous spread increases the chance of a persistent *L. monocytogenes* infection in the intestinal epithelium. (**A**) Schematic of the topology of the intestinal epithelium and *L. monocytogenes* cell-to-cell spread originating at the tip of intestinal villi. Image of intestinal villi was adapted from 'Structure of villi and microvilli present on the epithelial cells of the small intestine.' https://commons.wikimedia.org/
*Figure 5 continued on next page*

*Figure 5 continued*

wiki/File:Esquema_del_epitelio_del_intestino_delgado.png. License: CC BY-SA 4.0. Villus side view image was adapted from Figure 3 of *Pizarro-Cerdá et al. (2012)*. (B) Schematic of the steps of the host cell extrusion simulation. Host cell boundary of next cell to be extruded is depicted as a cyan dashed line. Extruding host cell is depicted as an orange boundary. Black arrows inside host cell in the second panel indicate monolayer retraction after extrusion. (C) Phase diagram of simulated data depicting the outcomes of host cell extrusion random walk simulations with only a single effective diffusion coefficient (no pioneers) for different combinations of host cell extrusion periods and effective diffusion coefficients. A total of 100 simulations were run for each combination. For these simulated data, the radius of extrusion equals 1, and the bacterial replication rate equals 1. Units are normalized by the replication rate, 0.006 min$^{-1}$, and radius of an extruded cell, 7 μm (*Ho et al., 2017*). The white dashed line represents the analytic approximation for extrusion steady state. (D) Simulated data showing the outcomes of host cell extrusion simulations where extrusion period equals 0.15, extrusion radius equals 1, $D_{slow}$ equals 2, and $D_{fast}$ varies. Vertical axis depicts the fraction of simulations with indicated outcome for 100 simulations per value of $D_{fast}/D_{slow}$ ratio.

DOI: https://doi.org/10.7554/eLife.40032.027

The following source data and figure supplements are available for figure 5:

**Source data 1.** This spreadsheet contains diffusion coefficient data used to generate the graph in *Figure 5C*.

DOI: https://doi.org/10.7554/eLife.40032.030

**Figure supplement 1.** Random walk host cell extrusion simulations predominantly lead to all-or-nothing outcomes.

DOI: https://doi.org/10.7554/eLife.40032.028

**Figure supplement 2.** Simulations predict that heterogeneous spread increases the chance of a persistent *L. monocytogenes* infection in the intestinal epithelium for several combinations of $D_{slow}$ and host cell extrusion periods.

DOI: https://doi.org/10.7554/eLife.40032.029

## Discussion

*Listeria monocytogenes* cell-to-cell spread has been primarily studied in two ways. First, plaque assays have been used to study late stages of infection where a few millions of bacteria have created plaques—sites of host cell death in cultured epithelial monolayers. The size of the plaque correlates to the efficiency of spread (*Van Langendonck et al., 1998*). Second, individual bacteria have been carefully observed by light and electron microscopy to provide information about the kinetics of protrusion formation and uptake (*Robbins et al., 1999*). In both cases, the identification of host and bacterial proteins has helped elucidate possible molecular mechanisms that facilitate *L. monocytogenes* spread (*Rajabian et al., 2009*; *Chong et al., 2011*; *Czuczman et al., 2014*). We were interested in bridging the gap between millions of bacteria creating millimeter-sized plaques and single bacteria creating micron-sized protrusions by studying cell-to-cell spread at a population level, while tracking individual bacteria at the frontier of the infection focus, with the goal of learning about both the collective and single-cell intercellular spreading behavior of *L. monocytogenes*.

We initially predicted that the spatial distribution of bacteria as a function of time would follow that of a random walk, a model characterized by isotropic, uncorrelated directions and normally distributed displacements (*Berg, 1993*). We developed this null hypothesis because (1) there is no evidence in the literature to suggest that intracellular *L. monocytogenes* motility has directionality, (2) late stages of *L. monocytogenes* cell-to-cell spread create circular plaques (*Van Langendonck et al., 1998*), and (3) MDCK cells form compact and relatively homogeneous monolayers in culture (*Mays et al., 1995*). Our high-resolution video microscopy assay, however, showed that a small number of

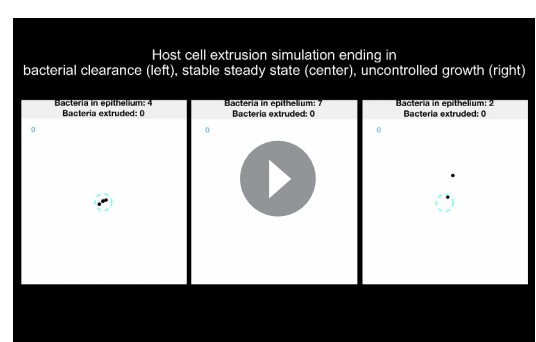

**Video 9.** Host cell extrusion simulation ending in bacterial clearance (left), stable steady state (center), and uncontrolled growth (right). Bacterial clearance is triggered when the number of bacteria in epithelium reaches zero. Uncontrolled growth is triggered when the number of bacteria in epithelium reaches $10^5$. Stable steady state is triggered when the number of extruded bacteria reaches $2 \times 10^5$ before the number of bacteria in epithelium reaches $10^5$. Extrusion radius equals 1, extrusion period equals 0.2, and replication rate equals 1. Effective diffusion coefficients are 1 (bacterial clearance), 3 (stable steady state), and 9 (uncontrolled growth ) respectively.

DOI: https://doi.org/10.7554/eLife.40032.031

bacteria spread farther than the rest and caused the infection focus boundary to become irregular. We propose that these pioneer bacteria spread by creating extracellular protrusions (*Pust et al., 2005*), that can reach and be taken up by recipient host cells that are not in direct contact with the donor cell.

Through simulations, we have found that allowing each bacterium to choose between two behaviors, far-reaching pioneer and local non-pioneer, approximated the shape of the experimental data better than simulating a single behavior of spreading bacteria (*Figure 3*). Simulated foci became less circular when the ratio $D_{fast}/D_{slow}$ was high and the total number of pioneer bacteria in the infection focus was very small (yet non-zero). While high numbers of pioneers are expected to increase the overall spreading rate of an infection, the anisotropic non-circular shapes of observed foci imply that the infection boundary is determined by rare events. Because pioneer bacteria themselves are assumed to not have a preferred spreading direction, small absolute numbers of pioneers are required in order to generate non-circular foci shapes as a result of stochastic fluctuations.

While we cannot rule out the possibility that infection focus anisotropy can also be attributed in part to host cell heterogeneity, the *L. monocytogenes* GRR mutant data suggest that properties of the bacteria themselves contribute significantly to this phenomenon. *L. monocytogenes* GRR mutants produce infection foci that are more circular than those produced by wild-type bacteria. Given that changing directional persistence in our simulations had little to no effect on circularity, but that changing $D_{fast}/D_{slow}$ did, it is likely that the GRR mutant *L. monocytogenes* creates shorter extracellular protrusions, that is GRR pioneers take shorter steps than wild-type pioneers. However, with our current experimental setup, we cannot determine whether these long protrusions seen in cultured cells (*Pust et al., 2005*) occur in the apical side of the monolayer, the basal side, or between cell-cell junctions in MDCK monolayers. Whereas at least a few of the pioneer events that we directly observed in MDCK cells appear to involve long apical protrusions, in the intestinal epithelium, which is characterized by a dense and highly organized apical brush border (*Crawley et al., 2014*), it is probably more likely that pioneers would spread either laterally between cells in the epithelium or basally at the junction between the enterocytes and the subjacent basement membrane. Indeed, *L. monocytogenes*' ability to cross the basal membrane of an epithelium via an actin-dependent process has been well-characterized (*Faralla et al., 2018*). The key feature of these protrusions, however, is that they enable an *L. monocytogenes* bacterium to bypass several host cells on its way to the more distant recipient cell. This model explains *L. monocytogenes*' ability to seemingly spread across two host cells in less than 30 min, even though the formation, uptake, and resolution of a single intercellular protrusion can take up to 45 min in MDCK cells (*Robbins et al., 1999*).

An extension of our model then argues that intracellular pathogens that spread from cell to cell without making extracellular protrusions would be expected to spread via a process resembling a random walk. The Gram-negative bacillus *Burkholderia thailandensis* is an example of a bacterial pathogen that spreads intercellularly primarily by inducing the cytoplasmic fusion of two neighboring host cells. Indeed, consistent with our pioneer model, infection foci created by *B. thailandensis* in mammalian host cells are significantly more isotropic than those created by wild-type *L. monocytogenes* (*French et al., 2011*).

This type of dual spreading behavior is not uncommon in other organisms. For example, the rice-water weevil, *Lissorhoptrus orzyzophilus,* migrates by both crawling and flying (*Shigesada et al., 1995*). If relatively few beetles migrate by flying, then early migration would be dominated by short steps and later migration would be dominated by longer steps. This ecological model parallels our heterogeneous *L. monocytogenes* cell-to-cell spread model given that: (1) the mean squared displacement accelerates with time in both rice-water weevil migration pattern data and *L. monocytogenes* cell-to-cell spread live microscopy assays (*Shigesada et al., 1995*), (2) the migration boundary of this organism deviates from a circle similar to bacterial infection foci (*Andow et al., 1990*), and (3) the rice-water weevil bimodal migration mechanism resembles that of bacterial local spread versus pioneer spread in long protrusions. In another ecological example, the population of European starlings, *Sturnus vulgaris*, is made up of short-distance and long-distance migrants. The latter of the two groups was able to establish colonies that helped to promote survival of the species (*Shigesada et al., 1995*). Indeed, we argue that heterogeneous spread increases the chance of *L. monocytogenes* survival in an actively extruding tissue (*Figure 5* and discussed below).

Dual spreading behavior can also occur in a variety of other microbial pathogens. Vaccinia virus undergoes intercellular cell-to-cell spread via an actin-mediated process resembling that of *L.*

*monocytogenes.* In addition, vaccinia virus can accelerate its own rate of spread by inducing the expression of viral proteins on the surface of an infected host cell. Upon encountering those proteins, new incoming viral particles are repelled from the already-infected host cells by actin projections and encouraged to infect virus-free host cells. The repulsion of superinfecting virions thus creates 'viral superspreaders,' whose spreading behavior resembles that of *L. monocytogenes* pioneers. Both viral superspreaders and bacterial pioneers skip host cells on their way to a recipient uninfected host cell, create anisotropic infection foci, and accelerate the pathogen's rate of spread (*Doceul et al., 2010*).

Even though mixing two random walks recapitulated the decrease in circularity seen in experimental data, we cannot rule out alternative spread models. For example, a well-characterized mathematical model is the Lévy flight, a random walk model where the step sizes are drawn from a heavy-tailed distribution instead of a normal distribution, thus ensuring a non-trivial fraction of arbitrary long steps (*Dubkov et al., 2008*). Lévy flights are used to model animals foraging for food: animals will take short steps as they are feeding and long steps as they are searching for the next feeding ground (*Viswanathan et al., 2008*). For *L. monocytogenes* cell-to-cell spread, a Lévy flight would indicate that at any given point, all bacteria have the ability to spread as either a pioneer or as a non-pioneer. With a Lévy flight, however, it is more difficult to mechanistically explain what allows a bacterium to become a pioneer. On the other hand, in our heterogeneous spread model, bacteria interconvert between two spread behaviors and retain that behavior until their next replication event. We designed the simulation this way to resemble a bacterium creating either a short protrusion or a straight, long protrusion, and replicating once they have broken out of the double-membrane vacuole in the cytoplasm of the recipient cell. The increase in directional persistence in pioneers is equivalent to a larger diffusion coefficient at long times.

Given that it is a foodborne pathogen, *L. monocytogenes* infections begin in the host's alimentary canal. In this work, we propose that *L. monocytogenes* may have evolved the ability to spread via host cell skipping to maximize its chances of surviving, replicating, and spreading in a host's actively extruding intestinal epithelium. Under the conditions set by our single random walk simulations, *L. monocytogenes*' ability to establish a stable steady state was attained by only a narrow set of effective diffusion coefficients (*Figure 5C*). Also, in this model, an individual effective diffusion coefficient usually led to an all-or-nothing outcome. If the step sizes were too small, then the infection was cleared 100% of the time. If the step sizes were too large, then the infection got out of control and caused uncontrolled growth 100% of the time. If the step sizes fell in a narrow range in between, the bacterium was able to successfully extrude many bacteria while sustaining the infection 100% of the time. This stable steady state is a desirable outcome for both bacteria and host: *L. monocytogenes* can promote the extrusion of its offspring, which can either (1) exit the animal via feces and infect other host animals, or (2) escape the extruded host cell and try to re-invade a different villus. This second point is important because our simulation considers a single villus only, even though the mammalian small intestine contains millions of individual villi (*Guyton and Hall, 2006*), each of which is a potential site of infection.

Given our findings, it was important to include pioneers in the host extrusion simulations. Simulating an effective diffusion coefficient that previously led to bacterial clearance and mixing it with larger effective diffusion coefficients allowed *L. monocytogenes* to attain a stable steady state (*Figure 5D*). Even though heterogeneous spread did not lead to 100% stable steady state, 10–15% of stable steady state infections become significant in the context of the millions of villi that make up the intestinal epithelium. Under these conditions, bacterial clearance was the most likely outcome, and uncontrolled growth remained low, unlike in the case of the random walk (*Figure 5C*). It is critical for bacteria to avoid uncontrolled growth in any single villus site as this could result in death of the host animal, which harms both the host and the pathogen since the pathogen can no longer replicate and spread to other hosts (*Falkow, 2006*). Importantly, high but not 100% of bacterial clearance allows *L. monocytogenes* to extrude more offspring while being able to achieve a stable steady state in a smaller fraction of villi. Finally, given that step size is a function of nutrient availability, temperature, and monolayer age, among other factors, our model predicts that heterogeneous spread widens the range of biological conditions that *L. monocytogenes* can explore to create a stable steady state. This is an import host-pathogen relationship because it does not harm the host and promotes pathogenic success. Beyond the gut, it is also possible that pioneers may be more successful at reaching distant organs within the host animal. In fact, it has been shown that a very small

number of founder *L. monocytogenes* bacteria can spread from the gut to organs such as the spleen and gall bladder, a process that leads to bottlenecking (*Zhang et al., 2017*).

In our current model of an actively extruding epithelium, host cell extrusion occurs at regular intervals and is not influenced by bacterial load. An alternate mechanism that would also be expected to lead to a stable steady state would be forcing extrusion to occur after a preset number of bacteria is reached. Uncontrolled growth is inhibited since bacteria are extruded as soon as the number gets too high, and bacterial clearance does not occur since extrusion stops if bacterial counts get low. However, this alternative mechanism would require that the host cell be able to sense the number of intracellular bacteria and specifically alter its behavior accordingly. Our model, in contrast, presents a simple physical mechanism by which steady state can be achieved without additional sensing capabilities on the part of the host cell.

In addition to *L. monocytogenes*, other pathogens have evolved strategies to create persistent infections in their hosts. For example, the lambda phage induces expression of the λ repressor to change its gene expression profile from an active host-killing lytic state to a dormant lysogenic state. During the lysogenic state, the lambda phage integrates its genome into the bacterial chromosome, which is then inconspicuously replicated by the host's DNA replication machinery (*Ptashne, 2006*). The lambda phage stays dormant until environmental conditions, such as host bacteria availability, indicate that it is safe to kill the donor and spread to recipient hosts. Just like pioneer *L. monocytogenes* behavior, spontaneous induction from a lysogenic to lytic state is rare, a characteristic that promotes phage replication (*Little et al., 1999*). A continuous lytic state, similar to *L. monocytogenes* taking large steps in an extruding monolayer, would cause indiscriminate host death, thus harming both host and pathogen. Indeed, strategies that help establish persistent infections are critical in creating stable host pathogen interactions that have evolved over millions of years.

## Materials and methods

**Key resources table**

| Reagent type (species) or resource | Designation | Source or reference | Identifiers | Additional information |
|---|---|---|---|---|
| Strain, strain background (*L. monocytogenes*) | 10403S; JAT607; wild-type + actAp:: mTagRFP | *Bishop and Hinrichs, 1987* PMID: 3114382 | | Strains were conjugated with actAp:: mTagRFP ORF as described previously (*Ortega et al., 2017*) |
| Strain, strain background (*L. monocytogenes*) | DP-L4032; JAT1348; GRR + actAp:: mTagRFP | *Skoble et al., 2001* PMID: 11581288 | | |
| Cell line (*C. familiaris*) | Madin-Darby canine kidney type II G cells | *Mays et al., 1995* PMID: 7657695 | RRID: CVCL_0424 | |
| chemical compound, drug | Alexa Fluor 488 phalloidin | | Thermo Fisher A12379; RRID:AB_2315147 | |
| Chemical compound, drug | Gentamicin sulfate | | MP Biomedicals 194530 | |
| Chemical compound, drug | Rat-tail collage type I | | Thermo Fisher A1048301 | |
| Software, algorithm | MATLAB | | Mathworks; RRID:SCR_001622 | Used image processing toolbox |
| Software, algorithm | Circularity measurement | *Zheng and Hryciw, 2015* DOI: 10.1680/geot.14.P.192 | Mathworks; RRID:SCR_001622 | Written in MATLAB |
| Other | DMEM low glucose, no sodium bicarbonate, no phenol red | | Sigma D5921 | |

*Continued on next page*

*Continued*

| Reagent type (species) or resource | Designation | Source or reference | Identifiers | Additional information |
|---|---|---|---|---|
| Other | Leibovitz's L-15 medium, no phenol red | | Thermo Fisher 21083027 | |
| Other | Foundation Fetal Bovine Serum | | Gemini Bio-Prod 900108 | Lot: A37C48A |
| Other | Phosphate buffered saline (PBS), no calcium, no magnesium | | Fisher SH30028FS | |
| Other | Brain heart infusion (BHI) | | BD 211059 | |

## Bacterial strains and growth conditions

All 10403S *Listeria monocytogenes* strains used in this study are summarized in *Table 1*. The plasmid pMP74RFP (*Ortega et al., 2017*) was stably integrated into the genome of GRR *L. monocytogenes* via conjugation with *E. coli* SM10 λpir as previously described (*Lauer et al., 2002*). Three days before carrying out infection assays, bacteria were streaked out onto BHI agar plates containing 200 μg/mL streptomycin and 7.5 μg/mL chloramphenicol. Bacteria were inoculated and grown in liquid cultures overnight as previously described (*Ortega et al., 2017*).

## Mammalian cell culture

Madin-Darby canine kidney (MDCK) type II G cells (*Mays et al., 1995*) were grown in DMEM with low glucose and no phenol red (Sigma D5921) and low sodium bicarbonate (1.0 g/L) in the presence of 10% fetal bovine serum (FBS) and 1% penicillin-streptomycin. For live microscopy assays, 24-well plastic-bottom plates (Ibidi 82406) were coated with 50 μg/mL rat-tail collagen-I (Thermo Fisher A1048301), diluted in 0.2 N acetic acid, for 2 hr at 37°C and air-dried for 24 hr. Wells were washed with DPBS once before seeding. MDCK cells were cultured and seeded as instant-confluent monolayers as previously described (*Ortega et al., 2017*).

## Infection assay

Flagellated bacteria (OD600 of 0.8) were washed twice with DPBS and diluted in DMEM. Host cells were washed once with DMEM, and bacteria were added at a multiplicity of infection (MOI) of 200–300 bacteria per host cell in a volume of 500 μL/well. Bacteria and host cells were incubated together at 37°C for 10 min. Host cells were washed three times with DMEM to remove non-adherent bacteria and were incubated at 37°C for 15–20 min to allow a small number of adherent bacteria to invade host cells. It was important to keep the number of invading bacteria low because it prevents foci from merging with others. Media was replaced for DMEM +10% FBS+50 μg/mL gentamicin, and host cells were incubated at 37°C for 20 min to kill adherent bacteria. Media was replaced for DMEM +10% FBS+10 μg/mL gentamicin, and host cells were incubated for approximately 4 hr. The total time starting with the three DMEM washes until the end of the incubation is 5 hr.

## Microscopy

For live microscopy assays, MDCK cells were cultured on rat-tail collagen-I-coated 24-well plates (Ibidi 82406) for 48 hr as described above. Five hours post-infection, host cells were washed with

**Table 1.** Bacterial strains expressing mTagRFP under the *actA* promoter used in this work.

| Genotype | Strain designation | Parental strain designation | Parental strain reference |
|---|---|---|---|
| Wild-type | JAT607 | 10403S | *Bishop and Hinrichs, 1987* |
| GRR | JAT1348 | DP-L4032 | *Skoble et al., 2001* |

DOI: https://doi.org/10.7554/eLife.40032.032

Leibowitz's L-15 once and incubated with 1 µg/mL Hoechst, diluted in L-15, for 10 min at 37°C. Cells were washed with L-15 three times and media was replaced with L-15 +10% FBS+10 µg/mL gentamicin. MDCK cells and *L. monocytogenes* were imaged every 5 min with a 20X air objective (NA = 0.75) in an inverted Eclipse Ti-E microscope using µManager's autofocus feature. Red channel (bacteria), blue channel (nuclei), and phase (MDCK monolayers) were imaged. Environmental chamber was equilibrated to 37°C for at least 2 hr prior to imaging.

For fixed microscopy assays, MDCK cells and *L. monocytogenes* were co-incubated for 22 hr after the addition of gentamicin. Host cells were washed once with DPBS and fixed with 4% formaldehyde for 10 min at room temperature. Paraformaldehyde was removed and quenched with 50 mM $NH_4Cl$ for 10 min. Membranes were permeabilized with 0.03% Triton-X100, diluted in DPBS, for 7 min. Samples were incubated with 0.2 µM AlexaFluor488 phalloidin, diluted in DPBS, for 20 min at room temperature.

## Image analysis

All image TIFF files were imported into MATLAB and processed with the image processing toolbox (MathWorks). To process experimental microscopy data, images were read in as 1024 × 1024 matrices, converted to double-precision numbers, and normalized to intensities ranging from 0 to 1. Images were thresholded using Otsu's method (*Sezgin and Sankur, 2004*). Bacterial debris was excluded from the thresholded mask by inspection.

To quantify the total fluorescence intensity for a given time point, the thresholded mask was dilated until the infection focus was represented as a single continuous round shape. The median of the intensity values found outside of the thresholded mask was set as the image's background, which was then subtracted from every value in the matrix. Finally, background-subtracted intensity values were summed. For a full time-lapse movie, total fluorescence intensity values were fit to an exponential function, which provided an estimated value for growth rate. Doubling time was calculated by dividing the natural log of 2 by the growth rate.

To quantify the mean squared displacement (MSD) for a given time point, the distance squared to each pixel of the thresholded mask was normalized by that pixel's fluorescence intensity. All normalized squared distances were averaged. For a full time-lapse movie, MSD values were fit to a linear function. The slope of this line was divided by four to estimate an effective diffusion coefficient.

To quantify the area of an intracellular bacterial focus for a given time point, the x y coordinates of the thresholded mask were calculated. MATLAB's boundary() function, using x y coordinates as input, was then used to calculate a boundary that fully encompasses all of the points while shrinking towards them. This function also returns the area contained inside the boundary. The radial speed of the focus is equivalent to the slope of the square root of the area divided by π plotted as a function of time (*Liebhold and Tobin, 2008*).

To quantify circularity, the boundary of the infection focus was used to calculate the smallest circle that fully encompasses the boundary, as described previously (*Zheng and Hryciw, 2015*). Then, the area of the boundary was divided by the area of a circle. A perfect circle thus has a circularity of 1.

To use the Voronoi tessellation to estimate the position of host cell boundaries, nuclei were thresholded as described above and segmented using a watershed transform. The center of mass of each nucleus was calculated and used as the input for MATLAB's Voronoi() function.

To calculate the *L. monocytogenes* point spread function (PSF), twenty 9 × 9 pixeled images containing individual bacterial cells, obtained from live microscopy experiments, were interpolated and aligned at subpixel resolution according to their center of mass. Images were averaged to create the PSF. Simulated data in Cartesian coordinates were binned in a 1024 × 1024 matrix and convolved with the PSF to generate data that matched the resolution of our microscope system.

## Simulation methodology

All data were generated from simulations written in MATLAB (*Ortega, 2018*; copy archived at https://github.com/elifesciences-publications/Listeria_spread_simulations).

At the beginning of the random walk simulations, several parameters are set: the effective diffusion coefficient (D), the replication rate (krep), the maximum bacteria to be accumulated (maxnbact), and the time-step (delt). Every run of the for loop is equivalent to a single time step during which (1)

bacteria age, (2) bacteria replicate, and (3) bacteria move. For bacterial aging, a vector called bacthist keeps track of each bacterium's age. These numbers increase monotonically until a particular number reaches that bacterium's replication time, drawn from a normal distribution with mean ln(2)/krep and variance mean/5. For replication, the positions of those bacteria whose age has reached their replication time are duplicated. Both daughters are assigned a new replication time from the same distribution and their ages are set to 0. Finally, the bacteria move in the x and y dimensions by sampling random numbers from the standard normal distribution scaled by the square root of 2*D*delt, where D is the effective diffusion coefficient and delt = 0.01 is the time step. At every time step, the number of bacteria, the MSD, the area of the boundary, and the circularity of the boundary, calculated as above, are recorded.

For heterogeneous spread simulations, the values of $D_{slow}$, $D_{fast}$, and P were set prior to the start of the simulation. P refers to the probability of becoming a pioneer. At the start of a bacterium's life, it chooses whether to spread according to $D_{slow}$ with probability P or according to $D_{fast}$ with probability 1–P.

To add persistence to the bacterial cell motility, two new parameters, θ and β, were included. In these simulations, the angle of movement is sampled from a normal distribution with mean θ (the angle associated with the previous step) and standard deviation β. For a random walk, β >> 2π, which means that any angle between 0 and 2π is equally possible. For a persistent random walk, β limits the angle of movement to values close to the angle of the previous step. When β = 0, the angle of movement is constant over time, and the bacteria will be perfectly persistent. To plot circularity as a function of persistence, one thousand random angles were generated for each value of β, and the cosine values of the angles were averaged. For β = 0, persistence was close to 1. As β increased, persistence was close to 0.

To add host cell boundaries, a Cartesian lattice was used to define boundaries between host cells. The parameter γ defines the probability with which simulated bacteria will cross the boundaries. In these simulations, the new bacterial positions are calculated, and those bacteria that do not cross a boundary are moved to the new positions. Those new positions that require boundary crossing are attained with probability γ. The remaining bacteria reflect from the boundary, remaining in the same cell.

For host cell extrusion simulations, a circular host cell (of size R = 1) is created in the center of the monolayer and extrudes after every fixed period of time. At this point, the simulated bacteria found inside the host cell were eliminated from the monolayer and cumulatively summed over the entire simulation. After extrusion, remaining bacteria are radially moved inwards by a distance equal to the radius of the extruded host cell. The number of bacteria at the beginning of the simulation is 100. If the number of bacteria goes to 0, then bacterial clearance is triggered. If the number of bacteria in the monolayer reaches $1 \times 10^5$, then uncontrolled growth is triggered. If the number of extruded bacteria reaches $2 \times 10^5$, stable steady state is triggered. Triggering any of these three outcomes causes the simulation to end.

Any combination of the above parameters (simple random walk, two effective diffusion coefficients, persistence, host cell boundaries, and host cell extrusion) can be used for any given simulation.

## Random walk theory

The reaction-diffusion equation we used to formalize the null hypothesis of an isotropic random walk is defined as follows:

$$\frac{\partial \phi}{\partial t} = D \frac{\partial^2 \phi}{\partial r^2} + k\phi \tag{1}$$

is a differential equation where Φ represents the bacterial concentration as a function of position and time, $t$ refers to time, $r$ refers to the position of the bacteria in polar coordinates or their radial distance to the center of mass, $D$ is the effective diffusion coefficient, and $k$ is the exponential growth rate. Its analytical solution (*Shigesada et al., 1995*) is:

$$\phi(r,t)=\frac{\phi_0}{4\pi Dt}\exp\left(\frac{-r^2}{4Dt}+kt\right) \tag{2}$$

where $\phi_0$ represents the initial concentration of bacteria located at the source (0,0).

To calculate the radial speed, that is how fast the focus grows after long periods of time, we set the above equation equal to some threshold concentration $\Phi$ and solved for the radial distance $r$ at which this threshold concentration is reached, as a function of time. We take the derivative with respect to time, and solve for the limit of dr/dt as time approaches infinity to obtain:

$$\lim_{t\to\infty}\frac{dr}{dt}=2\sqrt{Dk} \tag{3}$$

The step-by-step derivation has been previously described (*Andow et al., 1990*). *Equation 3* then predicts that stochastic simulations where D = 1 and k = 1 will generate infection foci that move a constant speed of 2 at long times (*Figure 2—figure supplement 2*).

We calculated the radial speed of simulated data by assuming that the area of the boundary is circular and thus can be approximated by $\pi r^2$. We divided the area of the boundary by $\pi$ and took the square root to obtain the average value of the radial distance, r, from the boundary to the origin of the simulation at (0,0). We plotted r as a function of time and took the slope of the linear fit to approximate dr/dt (*Liebhold and Tobin, 2008*).

## Analytic approximation for extrusion steady state

Here we derive an approximate relation between diffusivity and extrusion rate that yields a steady state in the case of bacteria spreading homogeneously as a random walk. The existence of a steady state requires that the rate at which new bacteria appear through replication equals the average rate at which bacteria are removed by extrusion, and that the spatial spreading of the focus in each extrusion period is balanced by contraction of the monolayer after removal of the extruded cell.

We assume that in steady state, the bacterial distribution just before an extrusion event can be approximated by a Gaussian distribution with variance $\sigma^2$. Each bacterium replicates at a rate $k_{rep}$, and is extruded at an effective rate $1/E*[1 - \exp(-R^2/\sigma^2)]$ corresponding to the extrusion rate times the probability of the bacterium being found within radius R of the center. In order for these two rates to be precisely balanced, we must have:

$$\sigma^2=\frac{R^2}{-\log(1-Ek_{rep})} \tag{4}$$

When a contraction operation corresponding to the extrusion event is performed on the steady-state Gaussian distribution, the new radial bacterial distribution is given by:

$$P(r)=\frac{1}{\mathcal{N}}(r+R)e^{-\frac{(r+R)^2}{\sigma^2}} \tag{5}$$

where $\mathcal{N}$ is a normalization constant. The mean squared radial displacement for such a distribution can be calculated as:

$$\langle r^2\rangle_{\text{post-ext}}=\sigma^2-\sqrt{\pi}R\sigma e^{\frac{R^2}{\sigma^2}}\text{erfc}\left(\frac{R}{\sigma}\right) \tag{6}$$

To achieve steady state, we must have $\langle r^2\rangle_{\text{post-ext}}$ +4 DE = $\sigma^2$, as additional spreading during the extrusion period should return the assumed variance $\sigma^2$ before the next extrusion event. This allows a solution for D in terms of $\sigma$ which, together with *Equation 4*, yields the following equality for maintaining steady state:

$$D=\frac{R^2\sqrt{\pi}\,\text{erfc}\left(\sqrt{-\log(1-Ek_{rep})}\right)}{4E(1-Ek_{rep})\sqrt{-\log(1-Ek_{rep})}} \tag{7}$$

## Calculation of fraction of bacteria with pioneer ancestors

The fraction of bacteria with at least one pioneer ancestor in their family tree can be calculated in a straight-forward manner by noting that each replication event in the chain of ancestors preceding

the bacterium resulted in a pioneer with probability P and that each of these choices of pioneer or non-pioneer identity are made independently of each other. Therefore, the probability that none of a bacterium's ancestors are pioneers is given by $(1 - P)^N$, where N is the number of generations preceding the bacterium. The probability of at least one pioneer ancestor is consequently:

$$1 - (1 - P)^N \qquad (8)$$

## Acknowledgements

We dedicate this work to Stanley Falkow (1934–2018).

We thank W James Nelson for providing MDCK type II G cells, and Alexander J Ball for wordsmithing. This work was funded by a Howard Hughes Medical Institute Gilliam Fellowship for Advanced Study and a Stanford Graduate Fellowship (FEO); a James S McDonnell Postdoctoral Fellowship Award in Complex Systems (EFK); and NIH Grant R37-AI036929 and the Howard Hughes Medical Institute (JAT).

## Additional information

### Funding

| Funder | Grant reference number | Author |
|---|---|---|
| Howard Hughes Medical Institute | | Fabian E Ortega<br>Julie A Theriot |
| James S. McDonnell Foundation | | Elena F Koslover |
| National Institute of Allergy and Infectious Diseases | R37 AI036929 | Julie A Theriot |

The funders had no role in study design, data collection and interpretation, or the decision to submit the work for publication.

### Author contributions

Fabian E Ortega, Conceptualization, Data curation, Software, Formal analysis, Validation, Investigation, Visualization, Methodology, Writing—original draft, Writing—review and editing; Elena F Koslover, Conceptualization, Formal analysis, Validation, Methodology, Writing—review and editing; Julie A Theriot, Conceptualization, Resources, Supervision, Funding acquisition, Methodology, Writing—original draft, Project administration, Writing—review and editing

### Author ORCIDs

Fabian E Ortega http://orcid.org/0000-0002-3782-4762
Elena F Koslover http://orcid.org/0000-0003-4139-9209
Julie A Theriot http://orcid.org/0000-0002-2334-2535

### Decision letter and Author response

Decision letter https://doi.org/10.7554/eLife.40032.035
Author response https://doi.org/10.7554/eLife.40032.036

## Additional files

### Supplementary files

• Transparent reporting form
DOI: https://doi.org/10.7554/eLife.40032.033

### Data availability

All data generated or analyzed during this study are included in the manuscript. MatLab code for simulations is available at https://github.com/Fabianjr90/Listeria_spread_simulations (copy archived at https://github.com/elifesciences-publications/Listeria_spread_simulations).

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
