## [Decision Letter]

[**Editorial note:** This article has been through an editorial process in which the authors decide how to respond to the issues raised during peer review. The Reviewing Editor's assessment is that all the issues have been addressed.]

Thank you for submitting your article "*Listeria monocytogenes* cell-to-cell spread in epithelia is heterogeneous and dominated by rare pioneer bacteria" for consideration by *eLife*. Your article has been reviewed by three peer reviewers, and the evaluation has been overseen by a Reviewing Editor and Wendy Garrett as the Senior Editor. The following individuals involved in review of your submission have agreed to reveal their identity: James Slauch (Reviewer #2); Babak Momeni (Reviewer #3). Reviewer #1 remains anonymous.

The Reviewing Editor has highlighted the concerns that require revision and/or responses, and we have included the separate reviews below for your consideration. If you have any questions, please do not hesitate to contact us.

Overall, this submission was well received by all three reviewers. The reviewers highlight that your work provides important insight into the infection process. All three reviewers feel that more explanation is needed to more clearly understand not only your experiments but also the conclusions that are drawn from them. We highly recommend that you address these issues brought up by all the reviewers when submitting your revised manuscript. This will allow the readers to fully understand and appreciate your scientific study on the concept of rare "pioneer" bacteria that are important for spread during an infection, in this case for Listeria.

Thank you for taking part in this form of peer review. We look forward to your revised manuscript.

Separate reviews (please respond to each point):

*Reviewer #1:*

I don't have any major concerns.

Minor Comments:

This is a very nice manuscript that details an interesting phenomenon that is likely to have impact on a number of both intracellular and extracellular pathogens. I only have a few minor comments.

1) The authors should note that the idea of a pioneer leading a charge that ends up in a different tissue site is a variation of their model, that leads to bottlenecking. In Listeria, this spreading of a few founder bacteria from the intestine into deeper tissue sites was first shown by Waldor and coworkers (Zhang (PMID: 28559314).

2) Figure 3D is really the heart of the paper and shows that the simulations have predictive value. Unfortunately, the measurement parameters of circularity have to be defined better within the manuscript for the nonmathematicians because this turns out to be the single most important measurement parameter in the manuscript. The concept is intuitive, but better definitions within the body of the results are necessary to explain what deviation from circularity means.

3) Subsection “Simulations predict that heterogeneous spread increases the chance of a persistent *Listeria monocytogenes* infection in the intestinal epithelium”: I don't like the term death to the animal, since during growth in tissue, death may occur at much lower loads of bacteria than what the authors predict for death. Uncontrolled tissue growth is more appropriate. Their definition of death is a blackening out of the movie images.

4) Figure 5D is quite important, but there is no definition of what the colors mean. I think that Red: Bacterial clearance; Green: host animal death; Blue: stable steady state, with blue stable when the Df/Ds ratio is stable

*Reviewer #2:*

Ortega et al. combine modeling and experimentation to promote the concept that rare "pioneer" bacteria, which spread beyond the neighboring host cells, are important in the biology of Listeria. This is an interesting and very well written paper that provides important insight into the infection process. It builds on quantitative knowledge gained over many years to inform their models. I have only minor comments on presentation.

Although generally clear, throughout the paper and in the figure legends, the authors should strive to acknowledge which parameters are based on assumptions and/or simplifications, and which are based on experimental data.

Minor Comments:

1) Figure 1C. Y axis. Average fluorescence intensity of what?

2) Subsection “Allowing simulated bacteria to interconvert between pioneer and non-pioneer behavior recapitulates the non-circular phenotype of experimental foci” and Figure 3D. As stated, the circularity of the simulations is dependent on time, but it is not clear how this time relates to the experimental results being used as the benchmark. Please comment.

3) Figure 5D. You need to label the lines. After I stared at it a little while, I realized that the "outcomes" colors are indicated under 5C, but it was not immediately obvious.

4) Subsection “Allowing simulated bacteria to interconvert between pioneer and non-pioneer behavior recapitulates the non-circular phenotype of experimental foci” and Figure 5C. You carefully discuss the fact that clearance versus death is dependent on the rate of host cell extrusion, but it is not clear what times you used for extrusion relative to replication, for example. Minutes, hours, days…?

5) I also wonder if you considered having extrusion dependent on the number of bacteria in the host cell instead of time.

6) Some people might read primarily the discussion. Redefine or just spell out "MSD" in paragraph five of the Discussion section.

Additional data files and statistical comments:

Seems more than thorough.

*Reviewer #3:*

The authors use modeling to infer the mechanism of cell-to-cell spread of *L. monocytogenes* in epithelial cells. They start by observing that despite expecting each infection to be clonal, the spread appears anisotropic. They then create simple models of random walk (continuum and agent-based) and note that their model predictions do not match experimental observations. Based on insights on known mechanisms of spread, they propose a different model in which a small fraction of progeny spread over longer distances, and examine whether such a model describes the observations properly.

The manuscript has a logical flow, is easy to follow, and has a nice combination of theoretical modeling and supporting experiments. It is generally well-written and contains useful and interesting information. I find the premise interesting and important, but I feel some of the results could benefit from additional explanation to clarify the rationale for some of the assumptions made in the model.

Main concerns:

1) One concern regards the construction of the model. The observed parameter that the authors have used to fit the parameters is "circularity", and their model they introduce two additional degrees of freedom (Dfast/Dslow and P), both of which affect circularity. When fitting the parameters of their model, they assume Dfast/Dslow = 100, and sweep over values of P (in Figure 3D) to find a P value for which their simulations match the experimental data. Since circularity is the only parameter they are using for fitting, it leaves the question open if with other values of these parameters the same outcome can be achieved (say, Dfast/Dslow = 10 and P = 1e-4). I suspect the authors have already examined this when constraining their model parameters, but I think it helps to explicitly mention their process of eliminating the alternative possibilities in their manuscript.

2) When examining the GRR mutant, the authors have mentioned that "This mutant… is less persistent than wild-type bacteria and it is therefore likely to enter protrusions at a lower frequency than wild-type bacteria and to form protrusions that are less straight". First, the authors should clarify what they mean by "less persistent", and why persistence matters.

Second, if I understand correctly, the reference they cite mentions that cell-to-cell spread is less efficient with GRR mutant and the trajectory of protrusions is less straight. I expected the authors to quantitatively assess the hypothesis that the spread of GRR mutant still follows their model (with different P and Dfast/Dslow parameters that they estimate). In my opinion, this is a missed opportunity in the current paper. The experiments are already done by the authors and there are some estimates of the difference between WT and GRR mutants in the reference they have cited. Thus, the only remaining part is making quantitative predictions based on the model about how the spread is expected to be and comparing those predictions with experimental data. If the results match, that would re-enforce the model, and if they don't, perhaps they can speculate what other factors might be involved.

3) In the last section of the Results: "Simulations predict that heterogeneous spread increases the chance of a persistent *Listeria monocytogenes* infection in the intestinal epithelium", in my opinion, there are aspects that need to be clarified. What is the significance of a stable steady state infection per infection site/villus? Based on the discussions in the paper, one can consider the overall infection as a metapopulation of several sites/villi. In this context, a steady-state infection would be important in the overall infection, not at each site/villus, since unstable sites/villi can still persist at the metapopulation level. This is because infections that spread too quickly are still primarily contained within that site (conceptually similar to coexistence with spatial refuge in a prey-predator ecological model) and the ones that are cleared can still infect other sites (dispersal-clearance balance, conceptually similar to mutation-selection balance in evolutionary theory). Without a more thorough investigation, it is not obvious to me why the special case of stable steady state would be the best strategy for maintaining an infection compared to these alternatives. I suggest de-emphasizing stable steady state infection at each site as the "evolutionarily preferred" solution for persistent infection.

Minor comments:

1) In determining the circularity, the authors mention that they use the smallest circle that contains all the points. How do you find this circle? Maybe I am overthinking it, but to me this is not trivial.

2) The choice of P = 0.01 is justified in Figure 3D, but it's not clear to me how the choice of Dfast/Dslow = 100 was made. Would you please elaborate?

3) Is the "effective" diffusion coefficient in Figure 5C the same as Dslow? I think it is and it helps to mention it explicitly.

4) I think it would have helped to include the spread for another microbe that does not have extracellular protrusions (no pioneer cells, as a point of reference), as a negative control. Would you observe a homogeneous plaque shape in these cases? If not, what are the parameters involved, and do they contribute to the heterogeneous spread observed for *L. monocytogenes*? I suggest this only as an optional addition, if such a system already exists.

---

## [Author Response]

Reviewer #1:

I don't have any major concerns.Minor Comments:This is a very nice manuscript that details an interesting phenomenon that is likely to have impact on a number of both intracellular and extracellular pathogens. I only have a few minor comments.1) The authors should note that the idea of a pioneer leading a charge that ends up in a different tissue site is a variation of their model, that leads to bottlenecking. In Listeria, this spreading of a few founder bacteria from the intestine into deeper tissue sites was first shown by Waldor and coworkers (Zhang (PMID: 28559314).

Thank you for the suggestion. We have added a few sentences about bottlenecking in paragraph ten of the Discussion and included a citation of Zhang et al., 2017.

2) Figure 3D is really the heart of the paper and shows that the simulations have predictive value. Unfortunately, the measurement parameters of circularity have to be defined better within the manuscript for the nonmathematicians because this turns out to be the single most important measurement parameter in the manuscript. The concept is intuitive, but better definitions within the body of the results are necessary to explain what deviation from circularity means.

We agree that we should have explained the concept of circularity better. In the Results section, we have added the following sentence: “For a perfect circle, this metric would be equal to 1, and for a square, this metric would be equal to 2/pi (Zheng et al., 2015).” This added citation also describes circularity in more detail.

3) Subsection “Simulations predict that heterogeneous spread increases the chance of a persistent Listeria monocytogenes infection in the intestinal epithelium”: I don't like the term death to the animal, since during growth in tissue, death may occur at much lower loads of bacteria than what the authors predict for death. Uncontrolled tissue growth is more appropriate. Their definition of death is a blackening out of the movie images.

This is a good point. We have changed all instances of “host animal death” to “uncontrolled growth.” We made these changes in the text; Figure 5 and Figure 5—figure supplement 1 (previously Figure S6A); and Video 9.

4) Figure 5D is quite important, but there is no definition of what the colors mean. I think that Red: Bacterial clearance; Green: host animal death; Blue: stable steady state, with blue stable when the Df/Ds ratio is stable

We apologize for the oversight. The figure legend is the same as Figure 5C. We have added the labels to Figure 5D.

Reviewer #2:

Ortega et al. combine modeling and experimentation to promote the concept that rare "pioneer" bacteria, which spread beyond the neighboring host cells, are important in the biology of Listeria. This is an interesting and very well written paper that provides important insight into the infection process. It builds on quantitative knowledge gained over many years to inform their models. I have only minor comments on presentation.Although generally clear, throughout the paper and in the figure legends, the authors should strive to acknowledge which parameters are based on assumptions and/or simplifications, and which are based on experimental data.Minor Comments:1) Figure 1C. Y axis. Average fluorescence intensity of what?

Thank you for catching this. This was meant to be “Total bacterial fluorescence intensity.” We have changed the axis label and figure legend accordingly.

2) Subsection “Allowing simulated bacteria to interconvert between pioneer and non-pioneer behavior recapitulates the non-circular phenotype of experimental foci” and Figure 3D. As stated, the circularity of the simulations is dependent on time, but it is not clear how this time relates to the experimental results being used as the benchmark. Please comment.

The times are comparable as 1 simulation step is, on average, equivalent to 1.7 minutes. Therefore, 800 steps, when simulations were stopped, are roughly equivalent to 1360 minutes, which is approximately the last time point of the experimental data. We have included this information in the figure legend.

3) Figure 5D. You need to label the lines. After I stared at it a little while, I realized that the "outcomes" colors are indicated under 5C, but it was not immediately obvious.

This was also noted by reviewer #1 (minor point 4). We have added the label to Figure 5D.

4) Subsection “Allowing simulated bacteria to interconvert between pioneer and non-pioneer behavior recapitulates the non-circular phenotype of experimental foci” and Figure 5C. You carefully discuss the fact that clearance versus death is dependent on the rate of host cell extrusion, but it is not clear what times you used for extrusion relative to replication, for example. Minutes, hours, days…?

We agree that it is useful to clarify the correspondence between times and distances in the simulation with real times and distances. The bacterial replication rate of 1 is equivalent to 0.006 replication events per minute. This was calculated by normalizing the simulation steps by the doubling time of intracellular *L. monocytogenes*, which is approximately 120 minutes in MDCK cells. Then, the period of host cell extrusion ranges from a period of 0.10 (i.e. 10 simulation steps) to 0.50 (i.e. 50 simulation steps). These translate to approximately 17 to 85 minutes respectively. The radius of a human enterocyte is approximately 7 µm (Ho et al., 2017), which scales the effective diffusion coefficients to speeds from 0 to 0.2 µm/sec, which is the average speed of intracellular *L. monocytogenes* (Robbins et al., 1999). We have added this information to the legend of Figure 5.

5) I also wonder if you considered having extrusion dependent on the number of bacteria in the host cell instead of time.

Thank you for raising this possibility. This kind of mechanism would be another way of enabling establishment of a stable steady state; by forcing extrusion to occur after a preset number of bacteria is reached, one effectively inhibits both uncontrolled growth (cannot get to high numbers since the host cell will then be immediately extruded) and bacterial clearance (since extrusion stops if bacterial counts get low). This scenario would require that the host cell be able to measure the number of intracellular bacteria in some way. Our model demonstrates an alternate physical mechanism by which steady state can be achieved without additional sensing capabilities on the part of the host cell. We have included a brief comment on this in the Discussion paragraph ten.

6) Some people might read primarily the discussion. Redefine or just spell out "MSD" in paragraph five of the Discussion section.

We have now fixed that.

Additional data files and statistical comments:Seems more than thorough.

Reviewer #3:

The authors use modeling to infer the mechanism of cell-to-cell spread of L. monocytogenes in epithelial cells. They start by observing that despite expecting each infection to be clonal, the spread appears anisotropic. They then create simple models of random walk (continuum and agent-based) and note that their model predictions do not match experimental observations. Based on insights on known mechanisms of spread, they propose a different model in which a small fraction of progeny spread over longer distances, and examine whether such a model describes the observations properly.The manuscript has a logical flow, is easy to follow, and has a nice combination of theoretical modeling and supporting experiments. It is generally well-written and contains useful and interesting information. I find the premise interesting and important, but I feel some of the results could benefit from additional explanation to clarify the rationale for some of the assumptions made in the model.Main concerns:1) One concern regards the construction of the model. The observed parameter that the authors have used to fit the parameters is "circularity", and their model they introduce two additional degrees of freedom (Dfast/Dslow and P), both of which affect circularity. When fitting the parameters of their model, they assume Dfast/Dslow = 100, and sweep over values of P (in Figure 3D) to find a P value for which their simulations match the experimental data. Since circularity is the only parameter they are using for fitting, it leaves the question open if with other values of these parameters the same outcome can be achieved (say, Dfast/Dslow = 10 and P = 1e-4). I suspect the authors have already examined this when constraining their model parameters, but I think it helps to explicitly mention their process of eliminating the alternative possibilities in their manuscript.

We included these data in what was previously Figure S5F (and has now been moved to Figure 3D). In this figure, we spanned a range of values of Dfast/Dslow and showed that larger values of Dfast/Dslow make foci less circular. We have expanded the text to make this more clear (subsection “Decreasing the persistence of bacterial motility leads to more circular infection foci”).

2) When examining the GRR mutant, the authors have mentioned that "This mutant… is less persistent than wild-type bacteria and it is therefore likely to enter protrusions at a lower frequency than wild-type bacteria and to form protrusions that are less straight". First, the authors should clarify what they mean by "less persistent", and why persistence matters.

We have expanded our explanation of directional persistence in subsection “Decreasing the persistence of bacterial motility leads to more circular infection foci”.

Second, if I understand correctly, the reference they cite mentions that cell-to-cell spread is less efficient with GRR mutant and the trajectory of protrusions is less straight. I expected the authors to quantitatively assess the hypothesis that the spread of GRR mutant still follows their model (with different P and Dfast/Dslow parameters that they estimate). In my opinion, this is a missed opportunity in the current paper. The experiments are already done by the authors and there are some estimates of the difference between WT and GRR mutants in the reference they have cited. Thus, the only remaining part is making quantitative predictions based on the model about how the spread is expected to be and comparing those predictions with experimental data. If the results match, that would re-enforce the model, and if they don't, perhaps they can speculate what other factors might be involved.

Thank you for this interesting suggestion. We ran additional simulations to test what feature of the change in behavior of the GRR mutant is most likely to allow the bacteria to make more circular foci. In these simulations, we varied both the directional persistence and the Dfast/Dslow ratio. The data show that persistence has little effect on circularity, particularly in the presence of pioneers (Dfast/Dslow > 1). Specifically, when bacteria are perfectly persistent, circularity actually increases in a random walk (at this artificial extreme, essentially all the bacteria simply move straight outward from the center). However, when pioneers are present, changing the persistence alone ends up having very little effect on circularity over a very large range of possible values. Given that changes in persistence over a degree quantitatively much greater than the actual persistence change for the GRR mutant has no effect on circularity in the simulations, but that changing Dfast/Dslow does, we conclude that the GRR mutant *Lm* likely is deficient in making long extracellular protrusions, i.e. GRR pioneers take shorter steps than wild-type pioneers. These new results are summarized in Figure 4—figure supplement 1 and discussed in more detail in the fourth paragraph of the Discussion.

3) In the last section of the Results: "Simulations predict that heterogeneous spread increases the chance of a persistent Listeria monocytogenes infection in the intestinal epithelium", in my opinion, there are aspects that need to be clarified. What is the significance of a stable steady state infection per infection site/villus? Based on the discussions in the paper, one can consider the overall infection as a metapopulation of several sites/villi. In this context, a steady-state infection would be important in the overall infection, not at each site/villus, since unstable sites/villi can still persist at the metapopulation level. This is because infections that spread too quickly are still primarily contained within that site (conceptually similar to coexistence with spatial refuge in a prey-predator ecological model) and the ones that are cleared can still infect other sites (dispersal-clearance balance, conceptually similar to mutation-selection balance in evolutionary theory). Without a more thorough investigation, it is not obvious to me why the special case of stable steady state would be the best strategy for maintaining an infection compared to these alternatives. I suggest de-emphasizing stable steady state infection at each site as the "evolutionarily preferred" solution for persistent infection.

This is an important point to clarify. At the level of the bacterial population infecting any single villus, it is most important for bacteria to avoid uncontrolled growth in any single site as this could result in tissue damage and possible death of the animal. But, if every infection site in the entire intestine were cleared, there would be no persistent infection to result in fecal shedding of replicated bacteria that can go on to infect other hosts. In this context, the observation that the “blue” range in Figure 5C is so narrow for the model based on a simple random walk suggests that it would be very difficult to hit this “sweet spot” for the whole metapopulation in the intestine, as the most likely outcomes at any given cell extrusion rate would always be either complete clearance at every site or rampant uncontrolled growth. At the level of the metapopulation, then, our results suggest that the “best” option for the bacteria to most likely persist at some few sites would be to have a fairly large range of parameters where both clearance (red) and stable steady state (blue) are possible outcomes for the bacterial population in any single villus. Figure 5D indicates that this is exactly the result of adding pioneer behavior to the cell extrusion simulations. So, having the expanded “blue” range in Figure 5D, i.e. an expanded regime to attain a stable steady state, means that not all bacteria were cleared and also that uncontrolled growth at any single site within the entire intestinal metapopulation could be avoided. This specifically is the outcome that we argue is most favorable for the bacteria. We have included a more complete explanation in paragraph eleven of the Discussion section.

Minor comments:1) In determining the circularity, the authors mention that they use the smallest circle that contains all the points. How do you find this circle? Maybe I am overthinking it, but to me this is not trivial.

Indeed you are right, this is not trivial. The detailed explanation for this algorithm can be found in Zheng and Hryciw, 2015, which we have now cited in subsection “Stochastic simulations of cell-to-cell spread via random walks are inconsistent with observed shapes of infection foci” and the Materials and methods section. In short, once the boundary of the focus is calculated using MATLAB’s boundary() function, the two farthest vertices are determined, which set the diameter of a temporary circle. If all the other points are contained within the circle, then the algorithm stops. If not, then the point that lies farthest outside of the circle is used to draw a new a circle (including the first two original points. The algorithm proceeds iteratively until the smallest circumscribing circle is determined.

2) The choice of P = 0.01 is justified in Figure 3D, but it's not clear to me how the choice of Dfast/Dslow = 100 was made. Would you please elaborate?

We chose the value of Dfast/Dslow to be 100 because the observed size of jumps for individual pioneer events are approximately 10 times longer than for non-pioneers, and the effective step size scales as the square root of the diffusion coefficient. We explain this in paragraph four of subsection “Allowing simulated bacteria to interconvert between pioneer and non-pioneer behavior

recapitulates the non-circular phenotype of experimental foci”.

3) Is the "effective" diffusion coefficient in Figure 5C the same as Dslow? I think it is and it helps to mention it explicitly.

Figure 5C describes a simple random walk with only a single effective diffusion coefficient. We have added “random walk simulations with only a single effective diffusion coefficient (no pioneers)” to the figure legend.

4) I think it would have helped to include the spread for another microbe that does not have extracellular protrusions (no pioneer cells, as a point of reference), as a negative control. Would you observe a homogeneous plaque shape in these cases? If not, what are the parameters involved, and do they contribute to the heterogeneous spread observed for L. monocytogenes? I suggest this only as an optional addition, if such a system already exists.

Thank you for the suggestion. Among bacterial pathogens that use this form of actin-based motility to spread from cell to cell, we do not know of any pathogens that are unable to make extracellular protrusions. However, in addition to being able to make protrusions (both intracellular and extracellular), *Burkholderia thailandesis* spreads primarily by inducing the fusion of neighboring host cells (French et al., 2011). If most spread occurs by fusion with neighboring cells, we would expect that this kind of spread would yield more circular foci than those dominated by pioneer behavior such as we have described here for *L. monocytogenes*. Indeed, *B. thailandesis* generates infectious foci that are indeed significantly more isotropic than those created by *L. monocytogenes* (French et al., 2011). We have added a new paragraph in the Discussion section describing this.